# Induction of anergic or regulatory tumor-specific CD4+ T cells in the tumor-draining lymph node

Ruby Alonso[1], Héloïse Flament[1], Sébastien Lemoine[2], Christine Sedlik[1], Emanuel Bottasso[1], Isabel Péguillet[3], Virginie Prémel[1], Jordan Denizeau[1], Marion Salou[1], Aurélie Darbois[1], Nicolás Gonzalo Núñez[1], Benoit Salomon[4], David Gross[2], Eliane Piaggio[1] & Olivier Lantz [1,3,5]

CD4+ T cell antitumor responses have mostly been studied in transplanted tumors expressing secreted model antigens (Ags), while most mutated proteins in human cancers are not secreted. The fate of Ag-specific CD4+ T cells recognizing a cytoplasmic Ag in mice bearing autochthonous tumors is still unclear. Here we show, using a genetically engineered lung adenocarcinoma mouse model, that naive tumor-specific CD4+ T cells are activated and proliferate in the tumor-draining lymph node (TdLN) but do not differentiate into effectors or accumulate in tumors. Instead, these CD4+ T cells are driven toward anergy or peripherally-induced Treg (pTreg) differentiation, from the early stage of tumor development. This bias toward immune suppression is restricted to the TdLN, and is maintained by Tregs enriched in the tumor Ag-specific cell population. Thus, tumors may enforce a dominant inhibition of the anti-tumor CD4 response in the TdLN by recapitulating peripheral self-tolerance mechanisms.

[1] Inserm U932, PSL University, Institut Curie, Paris 75005, France. [2] Inserm U1151, Institut Necker Enfants Malades, Paris 75015, France. [3] Laboratoire d'immunologie clinique, Institut Curie, Paris 75005, France. [4] Sorbonne Universités, UPMC Univ Paris 06, UMR-S CR7, Centre d'Immunologie et des Maladies Infectieuses (CIMI), INSERM U1135, CNRS ERL 8255 Paris, France. [5] Centre d'investigation Clinique en Biothérapie Gustave-Roussy Institut Curie (CIC-BT1428), Institut Curie, Paris 75005, France. These authors contributed equally: Ruby Alonso, Héloïse Flament, Sébastien Lemoine. Correspondence and requests for materials should be addressed to O.L. (email: olivier.lantz@curie.fr)

The T cells specific for tumor neoantigens (neoAgs), exclusively expressed by tumor cells, are not affected by central tolerance[1]. Although tumor neoAgs are often recognized by the immune system, tumors grow progressively in immunocompetent individuals[2]. The absence of clinically effective antitumor responses against tumor neoAgs may represent a particular case of peripheral tolerance. All the mechanisms that normally drive peripheral self-tolerance could be involved: deletion of T cells specific for neoAgs, immune deviation or suppression of the immune response[3–6]. In addition, tumors could initially be ignored in the absence of sufficient Ag in lymphoid organs[7], the only location to which naive T cells have access[8]. Therefore, tumor Ag-specific T cells would encounter their Ags when tumor burden is overwhelming[7]. Alternatively, tumor Ag-specific naive T cells might be primed in the tumor-draining lymph node (TdLN), but resistance and escape mechanisms within the tumor would prevent its destruction[9]. Thus, the respective impact of inefficient priming in the TdLN or resistance mechanisms in the tumor bed are not fully understood.

A lot of emphasis has been put to date on antitumor CD8[+] T cell response. CD4[+] T cells as direct mediators of antitumor responses are just beginning to be appreciated. CD4[+] T cells participate to tumor rejection by helping CD8[+] T cell priming or migration to the tumor bed, recruiting innate cells or directly killing tumor cells[10]. Accordingly, chronically activated effector CD4[+] T cell expansion and tumor regression are correlated during neo-adjuvant chemotherapy of patients with breast cancer[11]. Adoptive transfer of in vitro expanded tumor-specific autologous CD4[+] T cells can induce long-term complete remission in cancer patients[12,13]. On the contrary, CD4[+] T cells can also have protumoral effects through the immumodulatory capacity of Treg cells (Tregs). The number of Tregs is increased in the blood, TdLN and at the tumor site in mouse tumor models as well as in cancer patients. In addition, systemic or local depletion of Tregs can enhance antitumor immunity[14,15].

Several mechanisms can contribute to the increased number of Tregs found in cancer patients and mouse tumor models: recruitment/expansion of thymus-derived Tregs (tTregs) in the tumor site and/or the de novo generation of peripherally-induced Tregs (pTregs) within the tumor or TdLN. The respective contribution of these 2 susbsets have been seldom studied due to the lack of reliable markers to distinguish them[16]. tTregs recognizing self-Ags expand earlier and faster than effector T cells and inhibit the development of T cell responses against tumor-specific Ags[17,18]. Moreover, conversion of Ag-specific naive CD4[+] T cells into pTregs has been observed in two transplanted tumor models: a B-cell lymphoma expressing hemaglutinin A (HA), and a melanoma expressing ovalbumin (OVA)[19,20]. However, a lymphoma is in direct contact with the immune system since the earliest stage and OVA is in part secreted due to an internal signal sequence[21]. It is thus unclear whether pTregs specific for a non-secreted Ag expressed in slowly growing solid tumors may develop de novo from naive CD4[+] T cells.

Anergy of tumor Ag experienced CD4[+] T cells has also been evoked as a mechanism of immune tolerance[22,23] but its definition remained vague until recently. The expression of high level of FR4 and CD73 on FOXP3−CD44[hi] CD4[+] T cell represents a positive definition of anergic T cells[24]. These anergic T cells would also represent a pTreg precursor reservoir for the maintenance of peripheral self-tolerance[24]. So far, the presence and/or mechanisms of CD4[+] T cell anergy in the context of solid tumors have not yet been addressed.

Transplantation of tumor cells expressing a nominal Ag into mice is frequently used to study the way a tumor neoAg is recognized by naive T cells. However, these models do not recapitulate the slow growth of tumors in cancer patients and the large amounts of Ag released in an inflammatory context at the time of implantation artificially prime the immune system[25]. This caveat is particularly important for CD4[+] T cell responses because MHC-II peptide complexes can last in vivo for several weeks[25,26]. Thus, how the natural antitumor CD4[+] T cell response develops is still unclear. Genetically induced tumors represent more physiological models[27] but to our knowledge, they have not yet been used to study the CD4[+] T cell response against a cytoplasmic neoAg expressed by solid tumors, the most frequent situation in humans.

In this work, we use an improved lentivirus (LV) based genetically engineered lung adenocarcinoma model to characterize the fate of naive CD4[+] T cells following the recognition of a tumor-specific cytoplasmic Ag. The tumor Ag reaches the TdLN and activates naive CD4[+] T cells but the response is not efficient as the activated tumor Ag-specific CD4[+] T cells are anergized or converted into pTregs. Depletion of host Tregs and immunization at a distant site from the tumor indicates a key role for the TdLN in which Tregs locally inhibit the priming of tumor Ag-specific naive CD4[+] T cells in a dominant manner.

## Results

**Impaired antitumor CD4 response during tumor development.** We initially used a model of genetically engineered lung adenocarcinoma that had previously been described[28]: a bi-cistronic LV encoding both a Luciferase fused to an Ag chosen at will and a Cre recombinase is injected intratracheally (i.t.) into Kras[LSL-G12D/+]Trp53[flox/flox] (KP) mice. We appended the sequence of the MHC-II-restricted DBY Ag to the Luciferase as a model cytoplasmic tumor Ag. Adoptive transfer of DBY-specific TCR transgenic Marilyn T cells allows monitoring of tumor Ag-specific CD4[+] T cell response, mimicking naive T cells that have exited the thymus and encounter tumor Ags in the TdLN.

Because the LV may transduce Ag presenting cells (APCs), and the stability of MHC-II peptide complexes on mature dendritic cells (DCs) can last for several weeks in vivo[26], an immune response against DBY may be induced in the presence of virus-induced signals, at the time of inoculation, independently of Ag expression by tumor cells. Indeed, CFSE-labeled Marilyn cells transferred intravenously (i.v.) into KP, as well as in C57Bl/6 (B6) mice, soon after LV inoculation, were strongly activated and proliferated in the mediastinal (Med) dLNs (Supplementary Fig. 1a, b), suggesting that lung APCs are transduced by the LV. To reduce expression of DBY by APCs after virus inoculation, we inserted four tandem target sequences of the hematopoietic-specific mir142-3p after the Luciferase-DBY cassette to induce its degradation specifically in hematopoietic cells[29] (Supplementary Fig. 1c). Use of the modified LV (Mod-LV) successfully reduced the expression of Luciferase-DBY by the monocytic cell line U937, while it did not in the epithelial HEK-293LTV cells (Supplementary Fig. 1d). The activation and proliferation of Marilyn cells transferred into B6 mice receiving the Mod-LV were significantly reduced and became undetectable after 4 weeks (Supplementary Fig. 1e). The DBY-specific polyclonal endogenous CD4 response was also transient and limited. In the Med-dLN, the low number of endogenous DBY-specific CD4[+] T cells observed 9 days after Mod-LV inoculation virtually disappeared within 2 weeks. This early response was accompanied by a limited recirculation to the lung that did not increase over time and did not generate DBY-specific host Tregs (Supplementary Fig. 1f-h). Thus, APCs having captured DBY Ag from transduced cells or expressing residual Ag in the absence of complete silencing by the mir142-3p generate inefficient priming that rapidly vanishes without giving rise to Treg or memory cells.

The LV modification did not significantly change tumor growth (Supplementary Fig. 2a). The first tumors were detected by bioluminescence 10–12 weeks after inoculation with the Mod-LV whereas the tumors imaged at 20–24 weeks were tenfold bigger and often associated with metastasis in the TdLN (Med-dLNs). These stages are refered as "early stage" (ES) and "advanced stage" (AS) according to immunohistochemistry analyses (Supplementary Fig. 2b, c and Supplementary Data 1). Although lymphocytic aggregates were observed at all stages of tumor development, as previously described[30] (Supplementary Fig. 2d-f), bona fide tertiary lymphoid structures expressing PNAd (a marker of High Endothelial Venules) were only observed at the most advanced stages in the invaded thoracic wall (Supplementary Fig. 2d-iv-vii).

To assess the quality of the tumor Ag-specific CD4$^+$ T cell response during tumor development, CFSE-labeled Marilyn cells were transferred into KP mice bearing ES or AS tumors. Their activation and proliferation were analyzed 7 and 14 days later (Fig. 1a) in comparison with tumor-free mice inoculated i.t. with DBY peptide and CpG adjuvant. Marilyn cells proliferated in the TdLN at both stages of tumor development. The significant proliferation of Marilyn cells observed in the ES group indicates that a MHC-II-restricted tumor Ag is not ignored even in small tumors. However, the proliferation of Marilyn cells was lower in

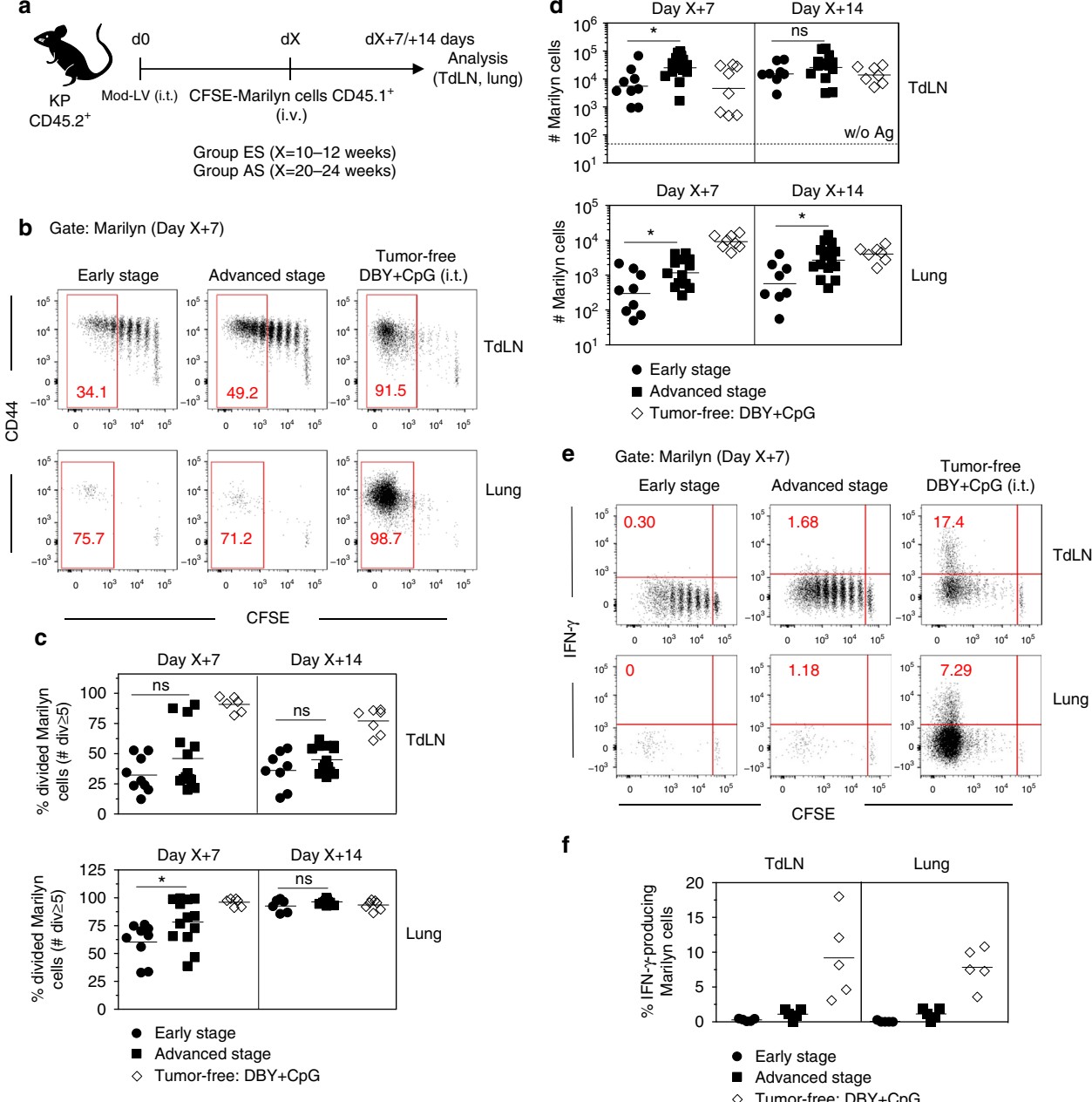

**Fig. 1** Priming of tumor Ag-specific naive CD4$^+$ T cells is not sufficient for full activation and migration to tumor site. **a** CFSE-labeled naive Marilyn cells were transferred into mice bearing ES or AS tumors. Their activation and proliferation were assessed 7 and 14 days after transfer in the TdLN and in the lung. **b**, **c** Pattern of proliferation and quantification (frequency of ≥5 divisions). Red gates indicate ≥5 cell divisions. Tumor-free B6 mice receiving 200 ng of DBY+CpG i.t. were used as controls. **d** Number of recovered Marilyn cells. Dashed line: basal number in the absence of Ag. Pooled data from three independent experiments. ns: non-significant, *$p < 0.05$ Mann–Whitney $U$ test. **e** Example of IFN-γ-producing Marilyn cell staining according to cell division after ex-vivo restimulation with PMA/Ionomycin for 4 h. **f** Quantification (frequency). One representative experiment out of three is depicted

both tumor groups in comparison to the DBY+CpG controls. In the TdLN, 7 or 14 days after transfer, <50% of Marilyn cells underwent ≥5 divisions in the tumor groups contrasting with 90% in the DBY+CpG group (Fig. 1b, c). In the lung, most of Marilyn cells were fully divided (Fig. 1b, c) but lung infiltration was strongly reduced as compared to controls at day 7 (Fig. 1d). The number of Marilyn cells increased slightly between the ES and AS in both the lung and the TdLN likely due to the higher tumor burden (Fig. 1d). As the pattern of Marilyn T cell proliferation was similar at both time points, we will thereafter focus on the day 7 after adoptive transfer.

After a short in vitro re-stimulation, only fully divided cells from the dLN and the lung of the control group produced IFN-γ (Fig. 1e, f). In line with our previous data[31], poorly divided Marilyn cells from tumor-bearing mice did not produce IFN-γ (Fig. 1e, f). Thus, tumor Ag reaches the TdLN in sufficient amount to induce activation and proliferation of tumor Ag-specific naive $CD4^+$ T cells, but the resulting priming is not efficient enough to induce full activation and migration of the tumor Ag-specific $CD4^+$ T cells to the tumor site, independently of tumor size.

**Tumor-specific pTregs arise early during tumor development.** We then characterized the tumor Ag-specific $CD4^+$ T cell response at ES and AS using gene expression profiling on total Marilyn cells from TdLN 7 days after transfer. Naive or activated Marilyn cells from tumor-free mice inoculated with male sple-nocytes or the Mod-LV were used as controls. Hierarchical clustering of most differentially expressed genes evidenced a set of genes specifically enriched in Marilyn cells activated in the tumor context. Interestingly, this cluster includes genes known as being expressed by Tregs, such as *Helios* and *Foxp3* (Fig. 2a and Supplementary Fig. 3). Notably, the whole transcriptome of activated Marilyn cells was identical in ES and or AS groups (Fig. 2b). A previously described canonical Treg signature[32] was enriched in both tumor groups in comparison with the LV control group (67.8% in ES and 64.4% in AS), as shown by the significant expression of canonical Treg genes such as *Foxp3*, *Il2ra*, *Nt5e*, and *Ebi3* (Supplementary Fig. 4a). Gene set enrichment analysis (GSEA) evidenced the presence of multiple Treg signatures in the Marilyn T cells from the ES and AS groups in comparison with the LV control group (Fig. 2c). Intracellular staining for FOXP3 confirmed the presence of Tregs in the TdLN 7 and 14 days after adoptive transfer. While FOXP3 expression could barely be detected in tumor-free mice receiving DBY+CpG, 3–5% of Marilyn cells expressed FOXP3 in the TdLN of both tumor groups. Similar numbers of $FOXP3^+$ Marilyn cells were found at ES and AS (Fig. 2d, e).

Importantly, Marilyn Tregs were derived from naive T cells since the transferred Marilyn cells were exclusively $CD62L^+CD44^{low}$ and did not express FOXP3 nor CD25 (Supplementary Fig. 4b). Peripheral induction of Tregs was further confirmed by transfer of FACS sorted $FOXP3-GFP^{Neg}$ Marilyn cells into tumor-bearing mice (Supplementary Fig. 4c, d). FOXP3 was observed only in Marilyn cells that were undivided or had divided 1-3 times in the TdLN and was barely detected in the >5 divided cells that had migrated into the lung (Supplementary Fig. 4e). Most of Marilyn pTregs expressed CD62L, PD1, GITR, CTLA-4, and Helios and some of them expressed CD25 and Nrp-1 (Supplementary Fig. 4f). Thus, when primed in the TdLN a significant proportion of naive $CD4^+$ T cells is rapidly converted into pTregs in both ES and AS tumors despite wide differences in Ag load between these two groups.

Although pTregs appear similar to tTregs with regard to gene expression and suppressive functions[33], the pTreg transcriptome

in a tumor context has not been characterized. To determine whether tumor-induced pTregs have a specific gene signature, we compared the transcriptomes of tumor-induced Marilyn pTregs and pTregs generated in tumor-free mice receiving i.t. DBY peptide alone (Supplementary Fig. 5) to natural Tregs (nTregs) from unmanipulated mice (Fig. 2f). 156 genes were differentially expressed in pTregs from tumor-free mice receiving DBY peptide as compared to 332 in ES and 498 in AS. 106 upregulated genes that were shared by ES and AS groups represent the tumor-induced pTreg signature (Fig. 2g and Supplementary Data 2) including genes associated with pTreg generation (*Pdcd1* and *Nfkbid*), Treg homeostasis (*Pdcd1*), and TGFβ sensitivity (*Tgfbr2* and *Tgfbr3*)[34–36]. Notably, pTregs generated by providing the DBY peptide in the absence of adjuvant in tumor-free mice were closer to nTregs than were pTregs generated in the tumor context suggesting that the latter may be more differentiated or activated.

**Tumor-specific $CD4^+$ T cells rapidly become anergic.** Although a significant fraction of tumor-Ag experienced Marilyn cells was converted into pTregs, most of them remained $FOXP3^{Neg}$. As pTregs can be generated together with $FOXP3^-CD44^{hi}CD73^{hi}FR4^{hi}$ anergic $CD4^+$ T cells[24], we looked for anergic Marilyn cells in TdLN and the lung of mice bearing ES and AS tumors. The vast majority of activated Marilyn cells in the TdLN displayed an anergic phenotype whereas an effector-memory (Teff-mem) phenotype ($FR4^{low/dim}CD73^{low/dim}$) predominated in Marilyn cells activated in tumor-free mice receiving DBY + CpG (Fig. 3a). The frequency of $CD73^{hi}FR4^{hi}$ anergic cells in the TdLN was higher at AS than at ES, reaching up to 90% of the $FOXP3^-CD44^{hi}$ Marilyn cells. Notably, most of the Marilyn cells that had accumulated in the lung displayed an anergic phenotype in both tumor groups (Fig. 3b). We then assessed cytokine production by the transferred Marilyn cells after in vitro restimulation with DBY Ag. Activated Marilyn cells in the tumor context did not produce IFN-γ or IL 10. Regarding IL-2, we did observe some level of secretion in the TdLN. However, the frequency of IL-2 + cells within the tumor was much lower than in the control (Fig. 3c, d). To better characterize the anergic Marilyn cells, we analyzed their transcriptome after isolation from the TdLN of mice bearing early or advanced tumors as compared to several control subsets. The anergic Marilyn cells were similar to Marilyn pTregs or to bulk Marilyn cells from our first data set (Fig. 2a), when compared to naive Marilyn cells or activated Marilyn cells from tumor-free mice receiving either the Mod-LV alone or immunized with DBY + CpG i.t. (Supplementary Fig. 6a). At the gene level, a very low number of genes were upregulated in anergic Marilyn cells (55 and 73 for early and advanced stage, respectively) when compared to pTregs. Conversely, the expression of a high number of genes was increased in pTregs (515 and 440 for early and advanced stage, respectively) when compared to the anergic cells (Supplementary Fig. 6b). These results suggest that pTregs and anergic Marilyn T cells share many characteristics, but anergic cells lack many important mediators of Treg functions such as *Foxp3*, *Il2ra*, and *Lrrc32* (GARP Glycoprotein A repetitions predominant) whose expression is highly decreased in anergic cells.

The anergic phenotype of $FR4^{hi}CD73^{hi}$ cells was further confirmed by a lower Ki67 expression. In the TdLN, anergic cells were $CFSE^{low}$ and few cells were $Ki67^+$ in contrast with Teff-mem cells from tumor-bearing mice or tumor-free mice receving DBY + CpG (Fig. 3e, f). The low Ki67 on anergic cells in tumor-bearing mice indicates that they have stopped to proliferate after several divisions (Fig. 3e) whereas most pTregs were undivided or had completed few divisions (Supplementary Fig. 4e). Thus, it is

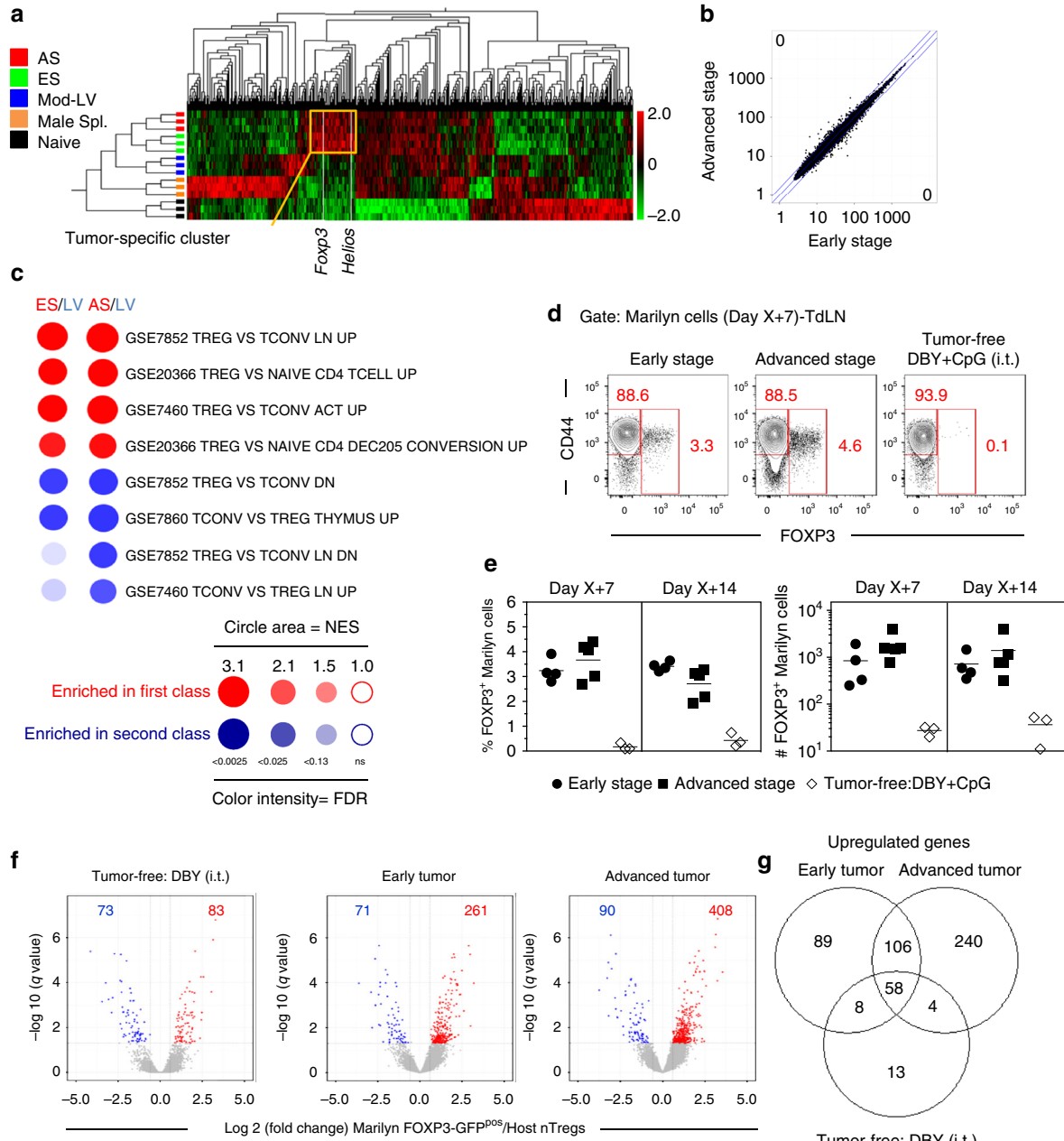

**Fig. 2** Activated tumor Ag-specific CD4$^+$ T cells display a Treg phenotype. **a–c** Gene expression profile of FACS-purified total Marilyn cells harvested 7 days after transfer from the TdLN of mice bearing ES or AS tumors. Controls are naive Marilyn cells (Naive) and activated Marilyn cells from tumor-free B6 mice primed with male splenocytes (Male Spl.) injected into the footpad (f.p.) or injected with the Mod-LV i.t. (Mod-LV). **a** Heat map and hierarchical clustering of the most differentially expressed genes ($q = 0.05$). The yellow square indicates the tumor-specific cluster. **b** Gene expression level comparison between Marilyn cells from mice bearing ES or AS tumors (no difference at $q = 0.05$). **c** Bubblemap of gene set enrichment analysis of ES vs. LV, or AS vs. LV conditions. NES, Normalized enrichment score, FDR, False discovery rate. **d** FOXP3 expression by Marilyn cells. **e** Frequency and number of FOXP3$^+$ Marilyn cells. Representative of one out of three independent experiments. **f** Volcano plots representing the q value against fold-change gene expression for FACS-purified Marilyn pTregs (FOXP3-GFP$^{pos}$) vs. polyclonal host nTregs from unmanipulated mice (CD4$^+$CD25$^{hi}$). Marilyn pTregs were purified from the Med-LNs of tumor-free mice receiving 200 ng of DBY i.t. or mice bearing early or advanced tumors, 7 days after the adoptive transfer of the naive cells. Upregulated or downregulated genes (Fold change>1.5, $q < 0.05$) are highlighted in red and blue, respectively. **g** Venn diagram of genes upregulated in Marilyn pTregs

unlikely that Marilyn pTregs derive from anergic cells. However, the anergic cells may derive from the effector/memory subset as in the TdLN, CD73$^{hi}$FR4$^{hi}$ anergic cells were more divided than the effector memory subset according to CFSE profile. Overall, the surface phenotype, cytokine production and proliferation profile of Marilyn T cells fit with the known features of anergic cells.

To further assess the filiation between the anergic and pTreg Marilyn cells, and to evaluate the stability of the anergic phenotype, CD44$^{hi}$FOXP3-GFP$^{Neg}$FR4$^{hi}$CD73$^{hi}$ Marilyn cells were purified from tumor-bearing mice and transferred into another set of tumor-bearing hosts (Fig. 4a). Fifteen days later, the transferred anergic cells were only found in the TdLN and not in the lung, inguinal/mesenteric LNs, spleen or bone marrow

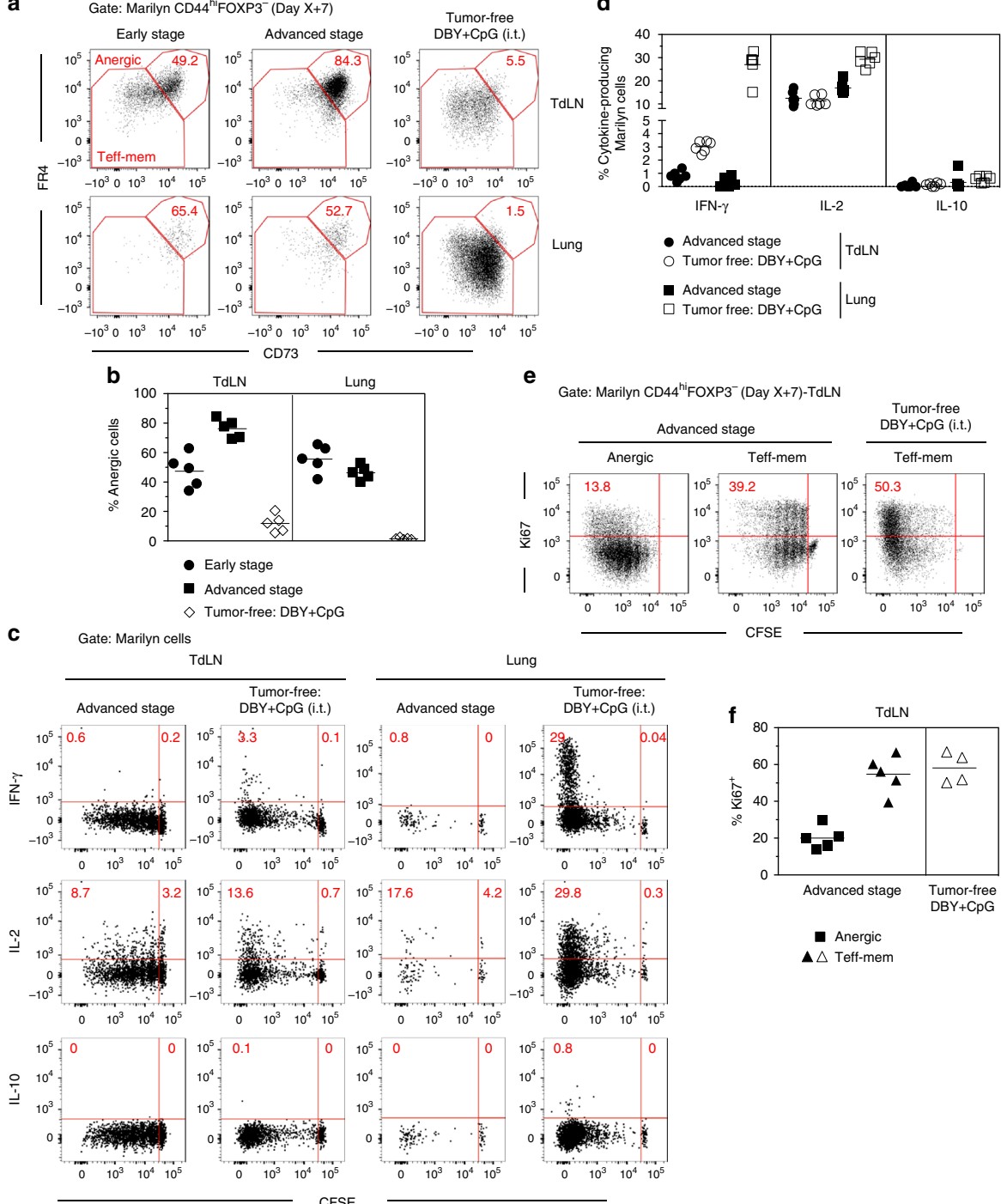

**Fig. 3** Most of activated tumor Ag-specific CD4$^+$ T cells becomes anergic. **a** CD73 and FR4 expression by CD44$^{hi}$FOXP3$^-$ Marilyn cells in tumor-bearing mice or tumor-free B6 mice inoculated with DBY + CpG (i.t.); **b** frequency of FR4$^{hi}$CD73$^{hi}$ (anergic) Marilyn cells. **c, d** Marilyn cells were harvested 7 days after transfer from TdLNs or lung of mice bearing advanced tumors. Cells were restimulated in vitro with CD3ε$^{-/-}$ female splenocytes pulsed with DBY peptide (10 nM). Controls were tumor-free mice i.t. injected with DBY + CpG. **c** Representative IFN-γ, IL-2 and IL-10 production by Marilyn cells according to cell division. **d** Quantification (frequency). **e** Representative Ki67 expression by anergic and effector cells in tumor-bearing or tumor-free mice. **f** Quantification (frequency). One representative experiment out of three is shown

(Fig. 4b). A low frequency (<20%) of transferred cells expressed Ki67 and all of them maintained the anergic phenotype (FR4$^{hi}$CD73$^{hi}$) without acquisition of FOXP3 (Fig. 4c, d). This formally demonstrates that anergic T cells do not convert into pTregs in this tumoral setting. However, after transfer into tumor-free B6 mice kept untreated or immunized with DBY

+CpG i.t. or f.p, former anergic cells were found in the lymph nodes draining the site of immunization and an important fraction also recirculated to the spleen and the lung (Fig. 4e, f). Moreover, up to 90% of Marilyn cells were cycling according to Ki67 expression (Fig. 4g) contrasting with the low proliferation observed in tumor-bearing mice (Fig. 4c, d). Importantly, the

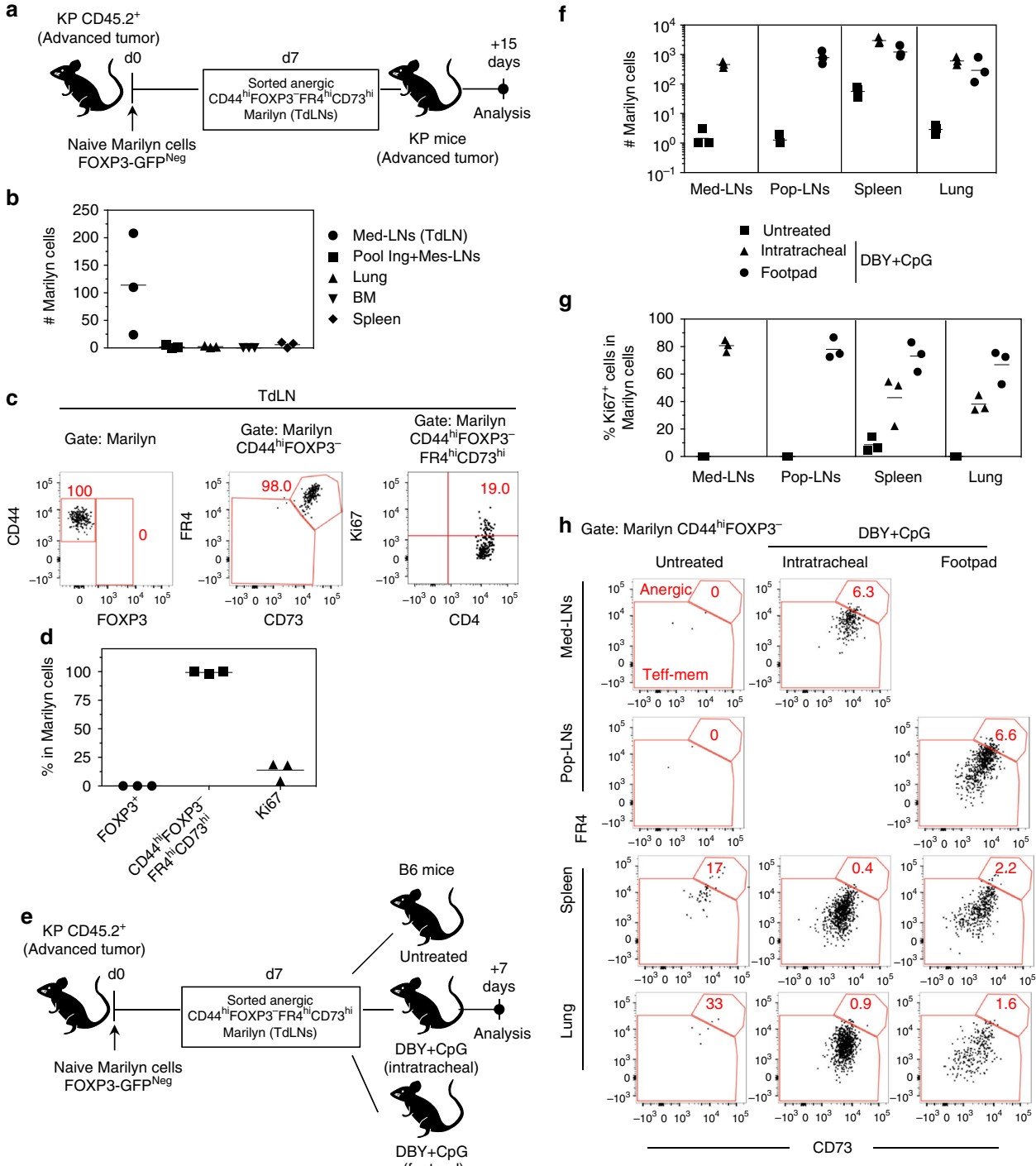

**Fig. 4** The persistence of the anergic phenotype requires Ag restimulation in the tumoral context. **a** FACS sorted CD44^hiFOXP3-GFP^negFR4^hiCD73^hi Marilyn cells from TdLN of mice bearing advanced tumors were transferred into a second cohort of tumor-bearing hosts. **b** Number of recovered Marilyn cells in the indicated organs (inguinal (Ing) or mesenteric (Mes) LNs, lung, bone-marrow (BM) and spleen) and **c**, **d** phenotype of Marilyn cells 15 days after transfer. **c** Representative dot plots. **d** Quantification (frequency). **e** FACS sorted CD44^hiFOXP3-GFP^negFR4^hiCD73^hi Marilyn cells from TdLN of mice bearing advanced tumors were transferred into tumor-free mice left untreated or treated i.t. or f.p. with DBY peptide and CpG. **f** Number of Marilyn cells 7 days after transfer in Med or poplietal (Pop) dLNs, spleen and lung. **g** Frequency of Ki67^+ cells among Marilyn cells. **h** Representative CD73 and FR4 plots

recovery of proliferation by the ex-anergic cells was associated with a loss of the FR4^hiCD73^hi anergic phenotype (Fig. 4h). Their loss was also observed in untreated B6 mice although to a lower level than in the Ag restimulated mice. Thus, tumor-induced CD4^+ T cell anergy requires continuous Ag presentation in a tumor context and can be reversed by Ag restimulation in the absence of tumor.

**TLR triggering does not rescue CD4^+ T cell functions**. We then studied the mechanisms leading to the generation of anergic and pTregs. Anergy induction and pTreg conversion observed in activated tumor-specific CD4^+ T cells could be the consequence of a lack of inflammatory signals in the tumor microenvironment leading to incomplete maturation of the APCs carrying the DBY Ag from the tumor to the TdLN. Intrapulmonary administration

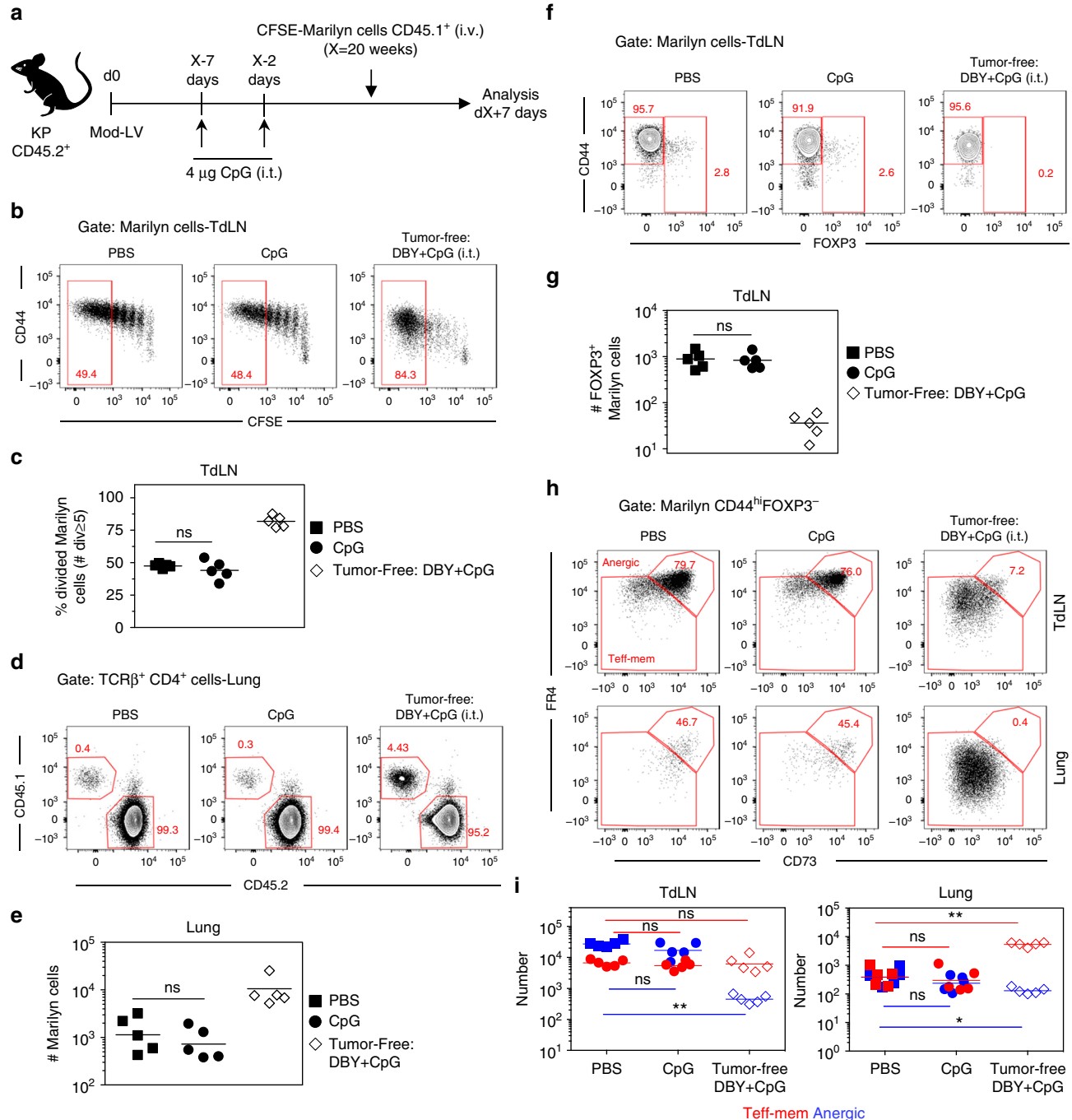

**Fig. 5** CpG administration at the tumor site does not restore an effector CD4[+] T cell response. **a** Mice bearing advanced tumors received 4 μg of CpG twice, at 5-day interval, 2 days before Marilyn cell transfer. **b** Proliferation profile of Marilyn cells in the TdLN of tumor-bearing mice receiving or not CpG. **c** Frequency of ≥5 times divided Marilyn cells. **d** Representative plots showing the frequency of Marilyn cells in the lung and **e** quantification (number). **f** Representative plots showing the frequency of FOXP3[+] Marilyn cells in the TdLN and **g** quantification (number). **h** Representative plots of FR4 and CD73 expression on CD44[hi]FOXP3[−] Marilyn cells, and **i** quantification (number of anergic or Teff-mem cells). ns: non-significant, *p<0.05, **p<0.01 Mann–Whitney U test. One representative experiment out of two is depicted

of the CpG adjuvant was sufficient to abolish pTreg induction in tumor-free mice receiving DBY peptide alone i.t. (Supplementary Fig. 5). We therefore injected CpG into the lung to induce optimal maturation of the APCs in mice bearing advanced tumors. To increase the likelihood that mature APCs would be present in the TdLN at the time of Marilyn T cell transfer, CpG was injected into the lung twice, 5 days apart, prior transferring CFSE-labeled Marilyn cells (Fig. 5a). In the lung, CpG administration induced the upregulation of CD86 by both CD11b[+] and CD103[+]

conventional DCs (cDCs) subsets in tumor-bearing mice. In the TdLN, both migratory and resident cDCs displayed increased expression of CD86 (Supplementary Fig. 7). However, Marilyn cell proliferation in the TdLN or their migration to the lung was not restored (Fig. 5b–e). Moreover, CpG administration did not prevent Marilyn pTreg induction in the TdLN (Fig. 5f, g) and did not significantly modify the frequency and number of anergic Marilyn cells in the TdLN and in the lung (Fig. 5h, i). Thus, either CD4[+] T cell unresponsiveness is not due to a lack of

inflammatory signal or tumor-related signals are dominant over the effect of CpG.

**Providing more tumor Ag does not restore CD4 response**. An efficient antitumor response requires that sufficient amount of tumor Ag reaches the TdLN. Non-secreted forms of tumor neoAg may only poorly stimulate specific naive CD4[+] T cells[37]. As Luciferase-DBY is a cytoplasmic Ag, we checked whether increasing availability of DBY at the tumor site would improve the CD4[+] T cell response. Tumor-bearing mice received DBY and CpG i.t., twice 5 days apart, before Marilyn transfer (Fig. 6a). Providing additional DBY Ag successfully restored Marilyn cell proliferation as well as migration to the lung (Fig. 6b–e). However, DBY+CpG administration led to an accumulation of Marilyn pTregs in the lung (Fig. 6f, g), without modifying the frequencies of anergic Marilyn cells in the TdLN and the lung (Fig. 6h). As the number of both Teff-mem and anergic cells increased in the same proportion, the ratio of Teff-mem/anergic cells remained unchanged in the TdLN and lung (Fig. 6i). Thus, increasing tumor Ag availability at the tumor site even in the presence of adjuvant expands both effector and suppressor cells.

**Host Tregs control anergy & pTreg conversion of CD4 T cells**. The frequency of host Tregs was higher in the TdLN and in the lung of tumor-bearing mice with a gradual increase from ES to AS (Fig. 7a, b). Gene expression profiling of host Tregs from mice bearing ES or AS tumors showed an enrichment of the recently described activated Treg signature[38] as compared to nTregs of unmanipulated mice (Fig. 7c). Many of the genes involved in Treg function were upregulated in Tregs from tumor-bearing mice (Fig. 7d). Importantly, in the TdLN and in the lung of mice bearing advanced tumors, host FOXP3[+] Tregs were enriched in T cells labeled with a DBY:I-A^b-tetramer (Tet). However, the frequency of this population in tumor non-draining LNs (pool of inguinal and mesenteric LNs) was comparable between tumor-bearing and tumor-free mice (Fig. 7e–g). Importantly, host DBY-specific Tregs were undetectable in the Med-LN of tumor-free mice whereas they were already present at early stage of tumor growth and subsequently accumulated during tumor development. At the tumor site, endogenous DBY-specific Tregs were found since the early stage without change in number at later time points (Supplementary Fig. 8). In line with these results, host Tregs from the TdLN and from the lung of mice bearing tumors but not from other locations were able to suppress Marilyn T cell proliferation in an in vitro Ag-specific suppression assay (Fig. 7h). These results suggest a potential role for tumor Ag-specific host Tregs in inducing de novo naive T cell conversion and the acquisition of an anergic phenotype.

To examine the role of host Tregs, we depleted these cells at the time of Marilyn T cell transfer using KP RAG2[−/−]/DEREG bone marrow chimera. Host Tregs were depleted in tumor-bearing mice by 3 diphtheria toxin (DT) administrations intraperitoneally (i.p.) at 2-day intervals. Naive Marilyn cells were transferred 1 day after the first DT dose (Fig. 8a). Following host Treg depletion the proliferation of Marilyn cells in the TdLN was significantly increased with 50–90% of the cells dividing ≥5 times (Fig. 8b, c) and was associated with higher infiltration in the lung (Fig. 8d, e). Host Treg depletion also significantly reduced the frequency and number of Marilyn pTregs in the TdLN (Fig. 8f, g), as well as the frequency of anergic cells both in the TdLN and the lung (Fig. 8h, i). DT administration reduced the number of anergic Marilyn cells in the TdLN to a level similar to that observed in the DBY +CpG control mice. On the contrary, in the lung, the number of anergic cells remained constant but was associated with a large increase of the Teff-mem pool (Fig. 8j). Therefore, in both

compartments the ratio of Teff-mem to anergic Marilyn cells turned in favor of the Teff-mem cells. Importantly, these results were the consequence of Treg depletion and not DT-associated inflammation as KP non-DEREG mice did not recapitulate the previous observations (Supplementary Fig. 9). Although activated host Tregs in tumor-bearing mice displayed an increased level of cell surface LAP (TGF-β complexed to latency-associated peptide) as compared to tumor-free mice, accompanied with GARP expression, TGF-β signaling was probably not involved in the de novo generation of tumor-specific Tregs or anergy induction (Supplementary Fig. 10a, b) since the administration of an anti-mouse TGF-β neutralizing antibody to mice bearing tumors did not modify Marilyn pTreg conversion or anergy induction (Supplementary Fig. 10c-e). Overall, these data suggest that host Tregs, enriched in tumor Ag-specific cells, generated/expanded during tumor development participate in the induction of anergy and the conversion of newly arriving tumor Ag-specific naive CD4[+] T cells into pTregs in the TdLN.

**TdLN is the specific site of tumor-induced tolerance**. These results suggest that the tumor establishes a state of dominant tolerance through the recruitment and generation of Tregs that hijack the priming of newly arriving Ag-specific CD4[+] T cells in the TdLN, ultimately affecting their accumulation at the tumor site. To confirm the predominant role of the TdLN in the induction of tolerance by the tumor, we set up a model where Ag-specific CD4[+] T cells are immunized in a distant LN (popliteal) by f.p. immunization with DBY+CpG in incomplete Freund's adjuvant (IFA) (Fig. 9a). Less than 50% of the Marilyn cells activated in the TdLN displayed a fully divided phenotype in unprimed mice. In contrast, all Marilyn cells from the lymph nodes draining the Ag injection site (IdLN) divided ≥7 times following immunization in tumor-bearing mice. In the TdLN of immunized tumor-bearing mice, the frequency of fully divided Marilyn cells was largely increased (Fig. 9b, c). This could reflect recirculation of activated Marilyn cells from the IdLNs or a positive feedback of those cells that would provide a pro-inflammatory environment ultimately promoting priming of CD4[+] T cells in the TdLN. The efficient priming of naive Marilyn cells in a non-tumor draining LN also led to a large increase of Marilyn cells in the lung (Fig. 9d, e). The priming at a distant site of the tumor did not modify the frequency of anergic Marilyn cells nor the ratio of Teff-mem/anergic cells in the TdLN. However, the immunization induced a large accumulation of Teff-mem Marilyn cells over anergic cells in the lung (Fig. 9f, g) and restored their IFN-γ production (Fig. 9h, i). Altogether, these results highlight the key role of the TdLN in establishing tolerance to tumor-specific Ags and the absence of systemic-specific immunosuppression.

**Discussion**

In this work, we demonstrate that tumors may set a dominant inhibitory environment in the TdLN, preventing efficient priming of tumor Ag-specific naive CD4[+] T cells that instead become anergic or pTregs. This effect is local and cannot be overcome by providing additional Ag and costimulation signals that would have efficiently primed otherwise in the absence of tumor. This effect is mediated at least in part by tumor Ag-specific host Tregs since their depletion leads to efficient priming and migration of effector CD4[+] T cells to the tumor. To our knowledge, our data provide the first comprehensive study of the CD4[+] T cell response toward a cytoplasmic tumor model neoAg expressed by newly transformed normal cells in a solid tumor. This is the first report of indisputable pTreg conversion and anergy induction of CD4[+] T cells in a tumor context. We also characterize in details

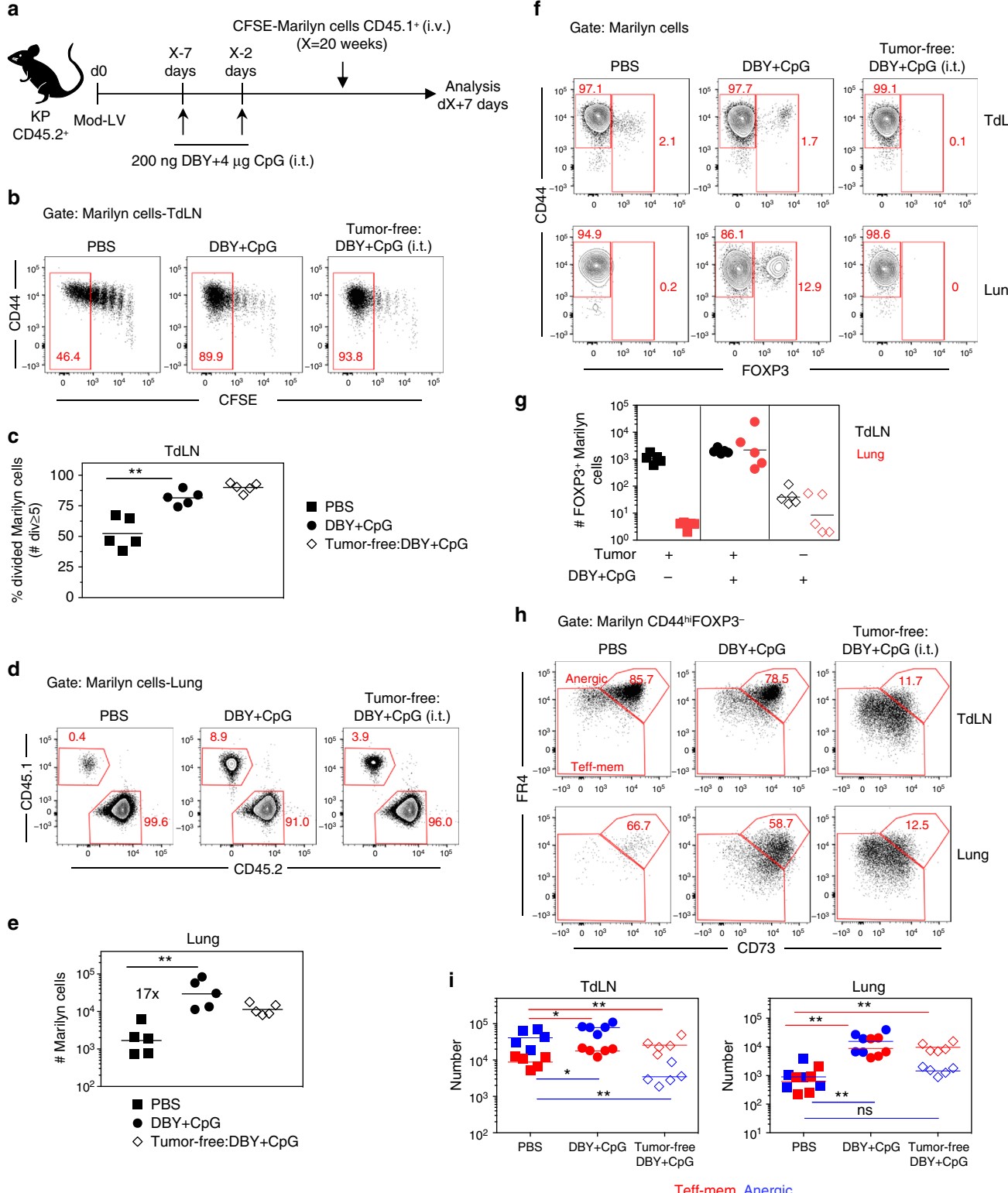

**Fig. 6** Concomitant administration of DBY and CpG at the tumor site reinforces tumor-mediated immunosuppression. **a** Tumor-bearing mice were i.t. injected twice with DBY peptide (200 ng) and CpG (4 μg) at 5-day interval followed 2 days after by CFSE-labeled naive Marilyn cell transfer. **b** Pattern of proliferation of Marilyn cells in the TdLN and **c** quantification (frequency of ≥5 divisions). **d** Representative plots showing the frequency of Marilyn cells in the lung and **e** quantification (number). **f** FOXP3 expression by Marilyn cells and **g** quantification (number). **h** Representative FR4 and CD73 expression on CD44^hiFOXP3^− Marilyn cells and **i** quantification (number of anergic or Teff-mem cells). ns: non-significant, *p<0.05, **p<0.01 Mann–Whitney U test

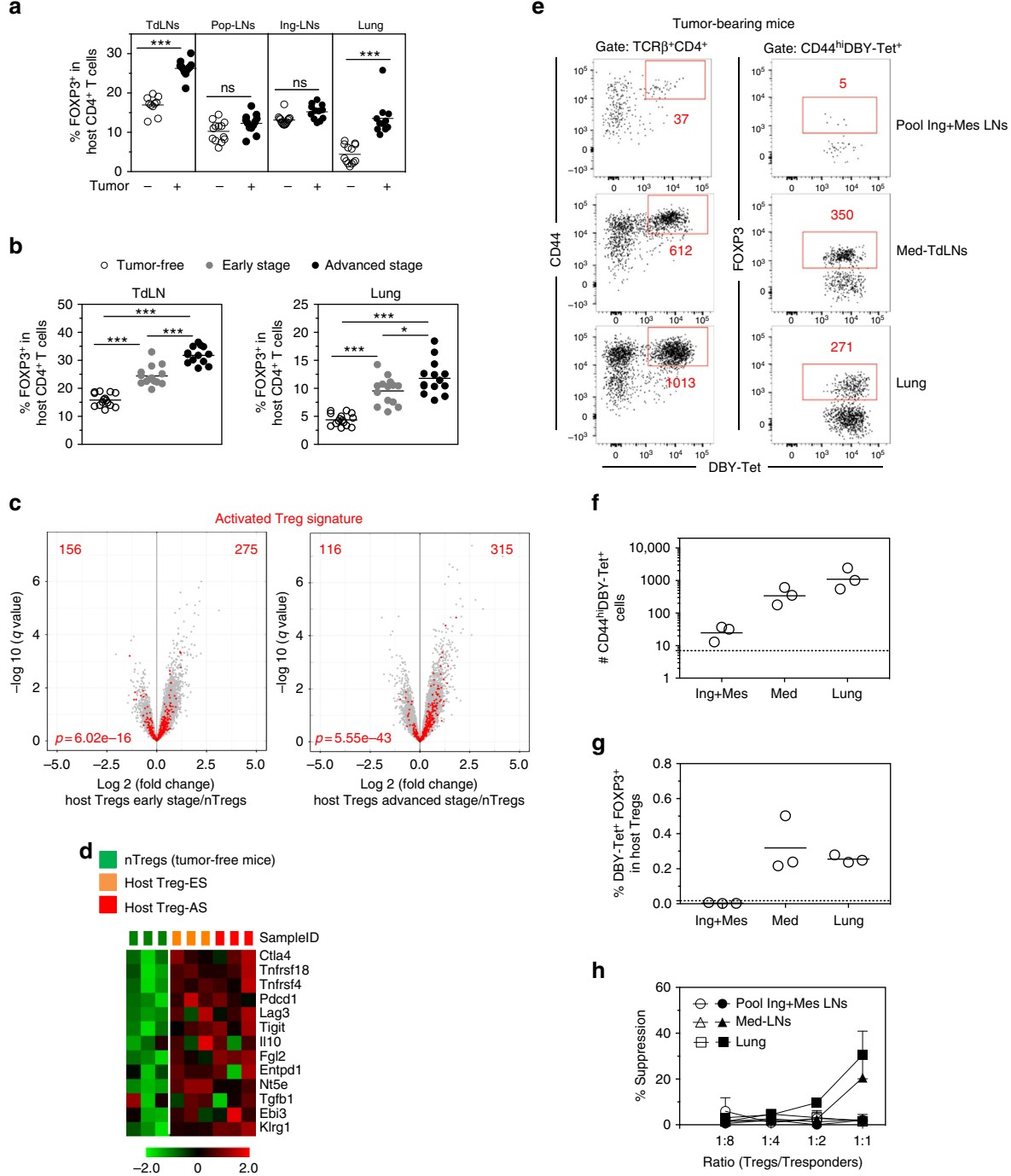

**Fig. 7** Host Tregs at the tumor site are enriched in tumor Ag-specific Tregs. **a** Frequency of host Tregs in the Med-TdLN, tumor non-draining LNs (popliteal = Pop-LNs or inguinal = Ing-LNs) and lung. **b** Frequency of host Tregs at different stages of tumor growth. ns: non-significant, *p<0.05, **p<0.01, ***p<0.001 unpaired t-test. **c**, **d** Transcriptional analysis of FACS-purified host Tregs (CD4$^+$CD25$^{hi}$) from Med-LNs of tumor-free or tumor-bearing mice. **c** Volcano plots comparing the q value vs. fold-change for host Tregs from mice bearing ES or AS tumors vs. host nTregs from unmanipulated tumor-free mice. Red dots represent the activated Treg signature. Numbers indicate the upregulated or downregulated genes among the specific signature. The enrichment p value is shown. **d** Heatmap of genes involved in Treg differentiation and function. **e** Representative plots of host CD4$^+$ T cells following DBY: I-A$^b$ tetramer (Tet)-based cell enrichment of cell suspensions from mice bearing advanced tumors. Total numbers of activated CD44$^{hi}$DBY:I-A$^b$-specific host CD4$^+$ T cells (left) and FOXP3$^+$ among DBY:I-A$^b$-specific host CD4$^+$ T cells (right) are shown. **f** Quantification of DBY:I-A$^b$-specific host CD4$^+$ T cells. Each point represents a pool of 2 mice. **g** Frequency of DBY:I-A$^b$-specific FOXP3$^+$ host CD4$^+$ T cells. Dashed lines represent the values obtained in LNs of tumor-free mice. **h** Suppression assay using purified host Tregs (FOXP3-GFP$^{pos}$) from tumor-free (open symbols) or tumor-bearing mice (closed symbols). Host Tregs sorted from Med-LNs, Ing-Mes-LNs or lungs were cultured for 3 days with CFSE-labeled naive Marilyn cells (Tresponders) and CD3ε$^{-/-}$ female splenocytes loaded with DBY peptide (2 nM). The graph represents the percent suppression of Marilyn cell proliferation (mean±SEM). One representative experiment out of two is depicted

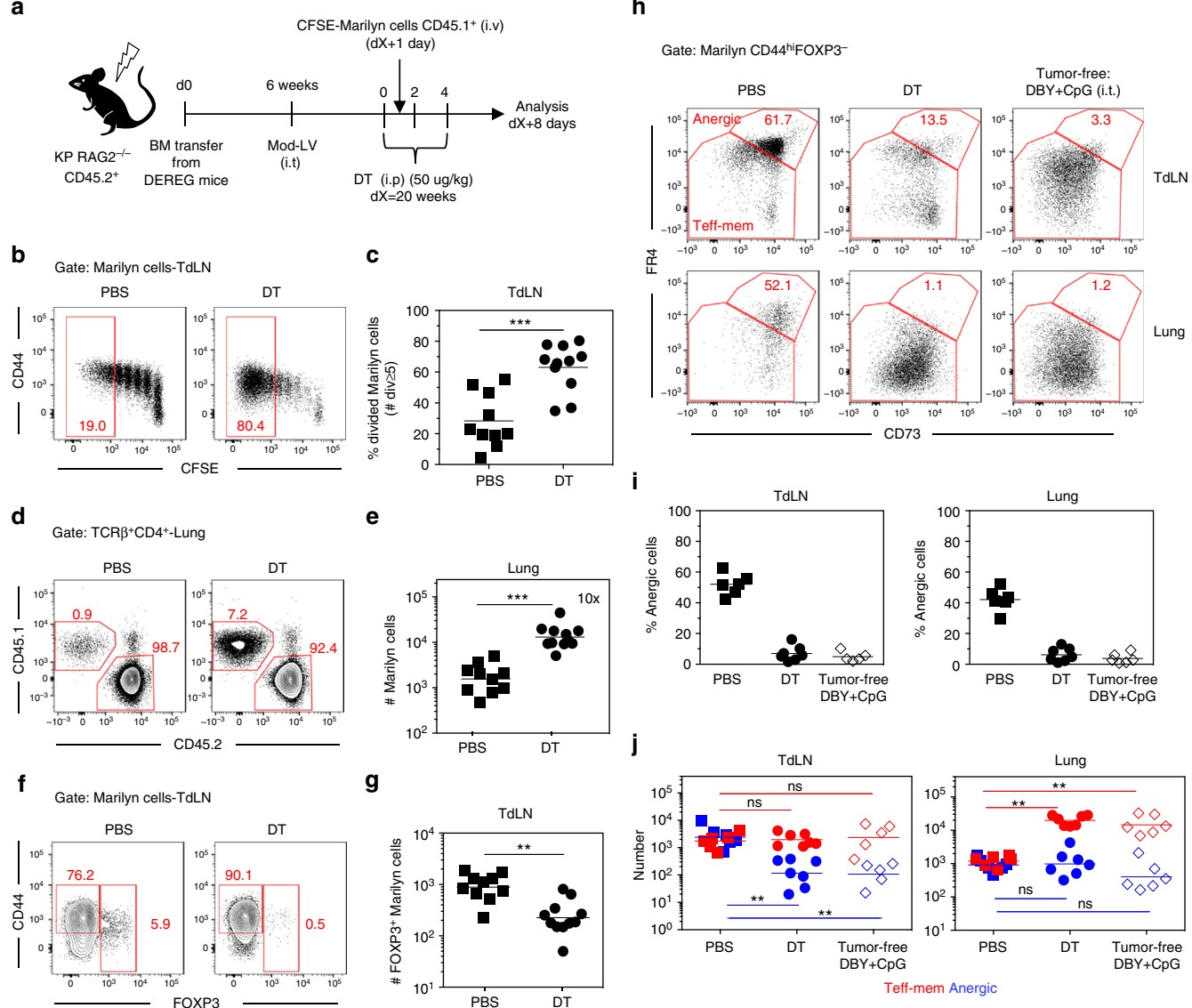

**Fig. 8** Depletion of host Tregs inhibits Marilyn T cell conversion into Tregs and restores their effector functions. **a** Irradiated KP RAG2$^{-/-}$ mice (5.5 Gy) were reconstituted with bone marrow from DEREG mice. Once the tumors were established, host Tregs were depleted by administration of 3 doses of DT i. p. at 2-day intervals. CFSE-labeled naive Marilyn cells were transferred 1 day after the first dose of DT and the analysis was performed 7 days later. **b** Pattern of proliferation of Marilyn cells in the TdLN and **c** quantification (frequency of ≥5 divisions). **d** Representative plots showing the frequency of Marilyn cells among the CD4$^+$ T cells in the lung and **e** quantification (number). **f** FOXP3 expression by Marilyn cells and **g** quantification (number). **h** Representative expression of CD73 and FR4 by CD44$^{hi}$FOXP3$^-$ Marilyn cells and **i**, **j** quantification (frequency and number). Pooled data of two independent experiments. ns: non-significant, **p<0.01, ***p<0.001 Mann–Whitney U test

these anergic CD4$^+$ T cells with regard to lymphokine secretion and phenotype stability in the presence or absence of tumor.

Tumor-induced CD4$^+$ T cell anergy has been proposed as an immune evasion mechanism in cancer[22,23]. In patients, the expression of inhibitory markers on tumor cells or APCs has been proposed to induce anergic T cells[39]. Our data expand the results obtained in the 90's bringing both a precise phenotype and a more physiologic context to the concept of tumor-induced anergy. Our data also suggest that tumor-induced anergy is not simply due to a lack of costimulation associated with an immature state of DCs as previously thought[40] but dependent on host Tregs in the TdLN as it was described for self-Ag-specific CD4$^+$ T cell anergy in a model of autoimmune arthritis[41]. Thus, in cancer, therapeutic manipulations targeting Tregs would not only release the brake from effector immune response but also impair

generation of anergic T cells, further enhancing antitumor immune responses.

Our data also provide insights regarding the filiation between tumor-induced pTregs, anergic and effector cells. Since Ag is present in the TdLN and not in other LNs, new naive CD4$^+$ T cells keep arriving at all time points in this asynchronous priming model, making lineage filiation considerations only tentative. Induction of FOXP3 in Marilyn cells was only observed in CD44$^{hi}$ T cells, indicating that activation is a prerequisite for Treg conversion. However, proliferation was not mandatory, as some FOXP3$^+$ T cells were undivided and many had divided only few times in accordance with previous reports[42,43]. This differs from the highly divided Tregs observed in a transplanted A20-HA lymphoma model[19]. This discrepancy could be related to the presence in this latter model of 5–10% of CD25$^+$ T cells in the

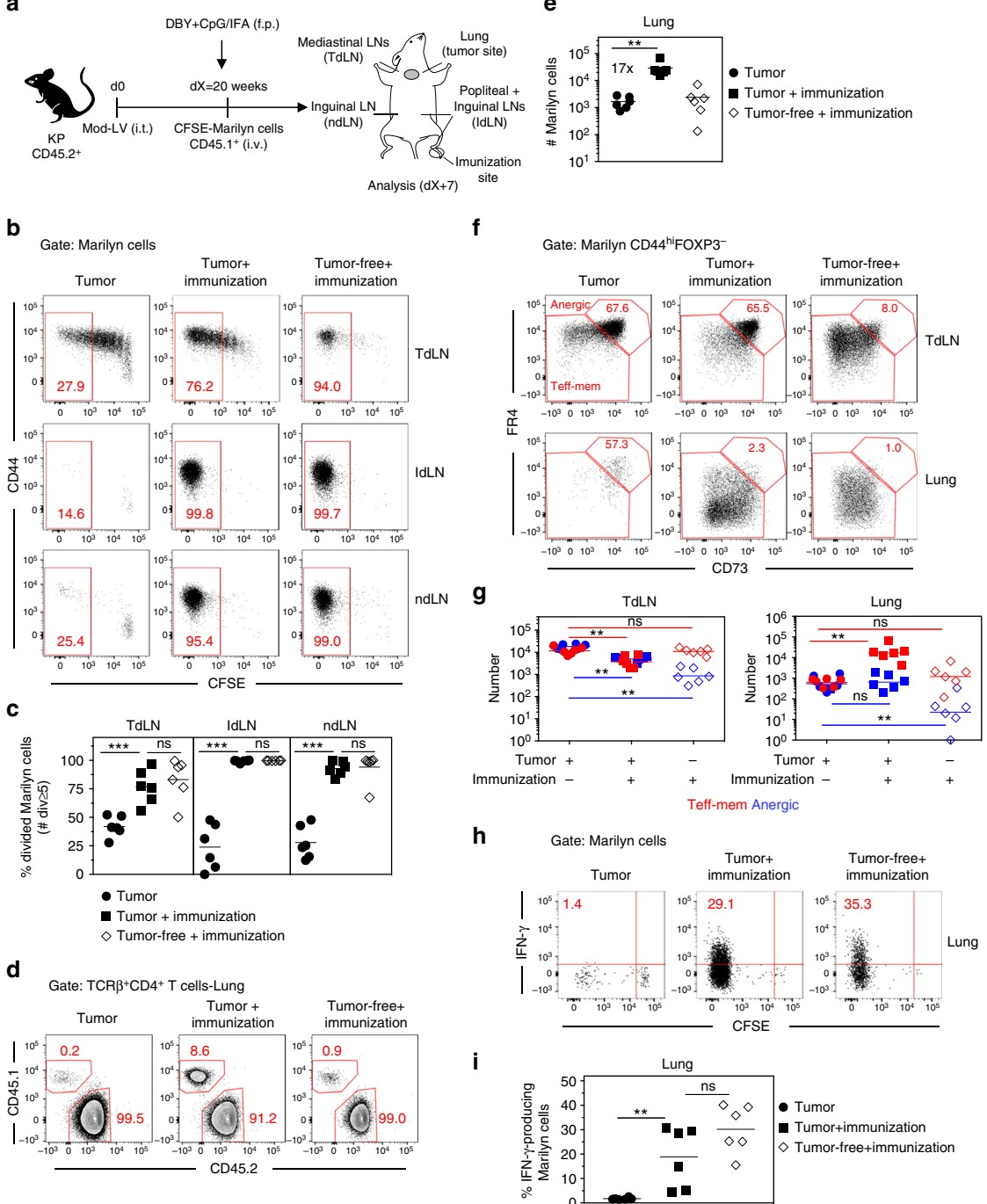

**Fig. 9** Priming of tumor Ag-specific naive CD4$^+$ T cells at distant site from the tumor restores their effector functions. **a** Tumor-bearing mice having received CFSE-labeled naive Marilyn cells were immunized into the f.p. with DBY peptide (200 ng) and CpG (40 μg) emulsified in IFA or left untreated. Analysis was performed 7 days later. **b** Pattern of proliferation of Marilyn cells in the TdLN, lymph nodes draining the Ag injection site (IdLN = popliteal + Inguinal-LNs) and ndLN (non-draining LN = Inguinal) and **c** quantification (frequency of ≥5 divisions). **d** Representative plots showing accumulation of Marilyn cells in the lung of immunized tumor-bearing mice and **e** quantification (number). **f** Representative expression of CD73 and FR4 by CD44$^{hi}$FOXP3$^-$ Marilyn cells and **g** quantification (number of anergic or Teff-mem cells) **h**, **i** Frequency of IFN-γ-producing Marilyn cells in the lung. One representative experiment out of two is depicted. ns: non-significant, **p<0.01, ***p<0.001 Mann–Whitney U test

transferred tumor-specific CD4$^+$ T cells, probably corresponding to tTregs. We used Rag2$^{-/-}$TCR transgenic mice that are devoid of any Tregs. In contrast to pTregs, anergic cells were only observed in the more divided subset suggesting that anergy does not precede pTreg conversion on the contrary to what has been described by Mueller's team[24]. This hypothesis was confirmed by

the absence of pTreg conversion after adoptive transfer of purified anergic cells in tumor-bearing mice. The origin of this discrepancy is still unclear. Altogether, our data suggest that tumor-induced pTregs are directly generated from recently activated naive cells and are not the progeny of the anergic cells, whereas the latter may result from effector cells.

The relevance of our finding is suggested by the presence of T cells similar to anergic cells in cancer patients. Our team recently evidenced that untreated metastatic uveal melanoma (mUM) and breast cancer patients harbor an increased frequency of $CD127^-CD25^-CD4^+$ T cells in the blood as compared to healthy donor[11]. This population of chronically stimulated $CD4^+$ T cells (chCD4) displays some of the features observed in mouse anergic cells: reduced proliferation and dampened cytokine secretion. We examined whether these chCD4$^+$ T cells would express a transcriptome signature similar to that of mouse anergic cells (Supplementary Fig. 11a, b and Supplementary Data 3). The tumor-induced anergic mouse gene signature was significantly enriched in chCD4$^+$ T cells from cancer patients ($p = 0.0023$) but not from healthy donors ($p = 0.275$) (Supplementary Fig. 11d), suggesting that $CD4^+$ T cells may also become anergic in tumor bearing patients including NSLC patients (Supplementary Fig. 11c). Thus, tumor-specific anergic $CD4^+$ T cells may also exist in humans.

With regard to the mechanisms of anergy and pTreg induction, depletion of host Tregs inhibited conversion of naive tumor Ag-specific $CD4^+$ T cells into pTregs and restored a typical effector phenotype, ultimately promoting their accumulation in the tumor bed. Whether host Tregs directly modulate naive Marilyn T cell priming through T–T interactions or modify the Ag-bearing APCs is unknown. Because in another model, secretion of the tumor Ag was necessary for tumor rejection by $CD4^+$ T cells alone[37], it is likely that the high number of $CD4^+$ effector cells generated by host Treg depletion will not be sufficient by themselves to reject the tumors in our model.

In contrast with the TdLN, the frequency of host Tregs in non-tumor dLNs was not increased. Notably, administration of DBY peptide with CpG at a distant site from the tumor but not at the tumor site led to strong Marilyn T cell activation with subsequent migration to the tumor site. These results demonstrate the key role of the TdLN in establishing a state of tolerance. The different frequency of host Tregs between TdLN and tumor ndLNs may account for the expansion of host Tregs-specific for tumor Ag(s) and/or self-Ag(s) from the organ where the tumor develops. The presence of DBY-specific Tregs in the TdLN and not in other LNs supports a role in anergy/pTreg conversion for tumor Ag-specific Tregs expanded/converted during the slow growth of the tumor.

Presumably, Treg mediated suppression whose primary function is to avoid organ-specific autoimmunity, also contribute to tumor tolerance. Accordingly, self-specific memory Tregs impair antitumor response by proliferating earlier than antitumor effector cells[17]. In our model, an early activation/expansion of host Tregs could occur at the stage of premalignant lesions as a normal host immune response to control autoimmune reactions. This Treg population presumably enriched in self and tumor Ag specificities would establish a state of tolerance avoiding the emergence of an effector response against tumor neoAgs. In a genetically induced prostate tumor model, Aire-dependent Tregs at the tumor site were mainly specific for a self-Ag[18]. However, in light of our results, tumor-specific pTregs also contribute to the enlarged Treg population found in the TdLN and at the tumor site.

Altogether, our data indicate that a non-secreted tumor Ag, that represents well most of the mutated proteins in cancer cells, is detected by tumor Ag-specific $CD4^+$ T cells. The responding $CD4^+$ T cells are not deleted but converted into anergic cells or pTregs. It will be important to better characterize and determine the life span of these subsets in humans as similar cells are detected in cancer patients. It is probable that the localized nature of the imprinting and the reversibility of the anergic state are dependent upon a priming or a boost in the absence of tumor Ag-specific Tregs. This opens the possibility of therapeutic actions.

Indeed, because in humans the thymic output is usually negligible in adults, the ability to therapeutically re-invigorate tumor Ag-specific $CD4^+$ T cells is necessary to use their direct antitumor properties and to avoid further downregulation of antitumor $CD8^+$ T cell responses due to an absence of help[44].

## Methods

**Mice.** Female $Kras^{LSL-G12D/+}Trp53^{flox/flox}$ mice (referred to as KP mice) generously provided by Dr. Tyler Jacks (Massachusetts Institute of Technology, Cambridge) had been backcrossed in house to a B6 background for >10 generations and used in this study as a genetically engineered mouse model of lung adenocarcinoma. They were used at 8 to 12 weeks of age. Female B6 Rag2$^{-/-}$ KP mice were used as recipients of bone marrow cell transfer. Around 8–12 weeks old female B6 CD45.1$^+$Rag2$^{-/-}$ Marilyn mice[45], whose transgenic TCR recognizes the I-A$^b$-restricted male DBY peptide ($NAGFNSNRANSSRSS$) were used as a source of monoclonal specific naive $CD4^+$ T cells. The three mouse strains described above were bred at CNRS Central Animal Facility (Orléans, France). Female B6 CD45.1/2$^+$Rag2$^{-/-}$ FOXP3$^{GFP}$ Marilyn mice were from the Inserm U1151 at Necker hospital. B6 FOXP3$^{DTR}$ mice coexpressing human diphtheria toxin (DT) receptor (DTR) plus GFP with FOXP3 (DEREG mice) were used as bone marrow cell donor mice[46]. Male B6 CD3ε$^{-/-}$ mice were the source of splenocytes harboring the DBY Ag and were bred in-house at Institut Curie. Wild type B6 mice were purchased from Charles River Laboratories (L'Abresles, France). All mice were housed in specific-pathogen free (SPF) conditions in full accordance with FELASA recommendations. All procedures had received approval from Institutional and Regional Ethical Review Boards.

**Patient samples.** Blood were obtained from metastatic uveal melanoma and non-small cell lung cancer patients before treatment in clinical trials IC-2004-01 and ALCINA IC-2015-02, respectively conducted at Curie Hospital. These studies were approved by the regional ethics board and all patients signed an informed consent form. Sodium-heparinized blood was collected and peripheral mononuclear cells (PBMCs) isolated using a standard Ficoll cell separation protocol. Healthy donor blood samples were obtained from the blood bank in accordance with French institutional regulations.

**Lentiviral particle production and titration.** A second-generation LV was produced by transfection of HEK-293LTV cell line (Cell Biolabs, Inc) with psPAX2 (packaging plasmid, Addgene), pCMV-VSV-G (envelope plasmid, Addgene) and Luciferase and recombinase Cre-expressing plasmid. The latter was generously provided by Dr. T. Jacks and modified in our laboratory to express the DBY epitope fused to the C-terminal end of Luciferase protein. Particles were purified through sucrose cushion as described[47]. Functional particles were quantified using a Cre activity readout system based on the reporter cell line 3TZ expressing -galactosidase upon Cre-mediated recombination.

**Preventing Luciferase-DBY expression in hematopoietic cells.** To restrict the expression of DBY to the non hematopoietic cell lineage[29], four tandem sequences of 23-bp complementary to the miRNA 142-3p (5′-TCCATAAAGTAGGAAA-CACTACA-3′), were inserted into the 3′untranslated region of Luciferase-DBY fusion protein cassette included in the Cre-expressing plasmid. The 4 × -mirT-142-3p were ligated to the previously digested Cre-expressing plasmid with NotI and XmaI enzymes.

**In vitro Luciferase signal detection.** U937 human monocytic cell line and HEK-293LTV human kidney cell line expressing or not miRNA 142-3p respectively[29], were transduced with serial dilutions of lentiviral particles. Luciferase expression was determined after 72 h in cellular lysates, using a Luciferase Reporter Assay (Promega) and expressed as relative light units (RLU) according to the manufacturer's instructions.

**In vivo tumor model and immunohistochemistry analyses (IHC).** Lung adenocarcinoma was induced in KP mice by intratracheal injection of $2 × 10^4$ lentiviral particles expressing the fusion protein Luciferase-DBY and Cre-recombinase as described[28]. In vivo tumor growth was measured by bioluminescence. Briefly, tumor-bearing mice were shaved and anesthetized with Isofluorane (Sigma-Aldrich). D-luciferin substrate (150 mg per kg, Promega) dissolved in PBS was injected i.p. 15 min before imaging in IVIS Spectrum (Perkin Elmer). Photon fluxes were transformed into pseudocolor images using the Living Image software (Perkin Elmer). Early and advanced tumor stages corresponding to 10–12 weeks and 20–24 weeks after lentiviral infection, respectively were validated by IHC analysis. Lung and TdLN were fixed in 4% paraformaldehyde, dehydrated and embedded in paraffin. Four µm sections were generated(Thermo Scientific Microtome) and used for further processing.

Sections were stained with Hematoxylin/Eosin, scanned (Philips Ultra Fast Scanner 1.6 RA) and digitally analyzed through a computerized image analyzer system (Philips Digital Pathology Solutions). The number of tumoral foci, their

area as well as the presence and number of atypical adenomatous hyperplasia (AAH) were quantified. For each tumoral focus the nuclear grade (Furhman nuclear grade) and the growth patterns were evaluated. Cell proliferation index was measured by Ki67 staining. The presence of metastasis in the TdLN was assessed by Epcam staining.

For IHC staining, 4 μm sections were boiled in citrate buffer (pH 6) or EDTA buffer (pH 9) for Ag retrieval in a pre-treatment module for tissue specimens (Dako). Sections were then treated with Serum-Free Protein Block (Dako) and incubated with primary antibodies. Peroxidase ABC system was used for signal amplification (Vectastain ABC Kit, Vector) and DAB detection system for revelation (Sigma-Aldrich) using manufacturer's specifications. For double staining CD3/B220, a secondary peroxidase-conjugated antibody was used (ImmPRESS Reagent Kit, Vector) to detect the CD3 staining combined with an AEC detection system (Peroxidase Substrate Kit, Vector). For B220 staining, a biotinylated secondary antibody was used and the signal amplified with Alkaline Phosphatase ABC system (Vectastain ABC Kit, Vector) and Vector Bleu detection system (Alkaline Phosphatase Kit III Substrate, Vector). Primary antibodies used are detailed in Supplementary Data 4. Slides were scanned and analyzed as above.

**Mice immunization and antibody treatment in vivo**. The DBY peptide was purchased from Genecust at 95% purity grade. Mice received 200 ng of DBY peptide alone, or DBY peptide plus 40 μg CpG-ODN 1018 (TriLink Biotechnologies) intratracheally (i.t.). When indicated, mice received twice 4 μg CpG i.t. at 5-day interval. For the experiment of priming at distant site of the tumor, tumor-bearing mice were immunized into the footpad (f.p.) with 200 ng of DBY peptide plus 40 μg CpG emulsified in 50% IFA (Sigma-Aldrich). For the transcriptome analysis, additional groups consisting in tumor-free B6 mice receiving $3 \times 10^6$ CD3ε$^{-/-}$ male splenocytes into the f.p. or DBY encoding lentiviral particles i.t. were used as controls. To neutralize TGF-β in vivo, tumor-bearing mice received 4 i.p. injections (5–10 mg per kg) every 3 days of the mouse monoclonal 1D11.16.8 (anti-mouse TGF-β, IgG1, BioXCell).

**Adoptive T cell transfer**. Naive CD4$^+$ T cells were extracted from a pool of lymph nodes of female Marilyn mice. Marilyn cells were labeled with 5μM CFSE (Carboxyfluorescein succinimidyl ester, Life Technologies) in PBS containing 0.1% BSA (Bovine Serum Albumin, Sigma-Aldrich) for 8 min at 37 °C. $2.5 \times 10^5$ cells were injected intravenously (i.v.) in 100 μl PBS 0.1% BSA in mice bearing early or advanced tumor. Analyses were performed 7 or 14 days after adoptive transfer. Anergic Marilyn cells (CD44$^{hi}$FOXP3-GFP$^{Neg}$FR4$^{hi}$CD73$^{hi}$) were purified from the TdLN of mice bearing advanced tumors and $3 \times 10^4$ anergic cells retransferred in new chohorts of tumor-bearing or tumor-free hosts. Mice were left untreated or immunized with DBY + CpG i.t or into the f.p.

**In vivo host Treg depletion**. FOXP3$^{DTR}$ KP mice were generated by bone marrow reconstitution. Briefly, Rag2$^{-/-}$ KP mice were irradiated (single exposure of 5.5 Gy in a Philips MG325 X ray generator) and 4 h later, $2 \times 10^6$ CD3-depleted bone marrow cells from B6 FOXP3$^{DTR}$ (DEREG) mice were transferred i.v. Reconstitution was confirmed in blood samples 4 weeks later. Tumor induction was triggered 6 weeks after reconstitution by LV administration. Tregs were depleted in mice bearing advanced tumor by i.p. injections of 50 μg per kg of DT (Merck Germany) dissolved in PBS. Three doses of DT were injected every two days and $2.5 \times 10^5$ naive CD4$^+$ Marilyn cells were transferred i.v. 1 day after the first dose, corresponding with the optimal depletion of host Tregs. Depletion efficiency was checked in blood samples at several time points.

**Mouse cell preparation**. TdLN were collected into CO$_2$ Independent Medium (Life Technologies). Single cell suspensions were obtained by mechanical disruption over a 40 μm cell strainer. For myeloid cell analysis, TdLN were digested with 0.1 mg/ml Liberase TL (Roche) in the presence of 0.1 mg/ml DNase (Roche) for 30 min before the addition of PBS supplemented with 0.5% BSA and 10 mM EDTA (Sigma-Aldrich) followed by mechanical disruption. Lungs were perfused with 20 ml of cold PBS to remove circulating blood cells and lung perfusion efficacy was assessed by intravascular staining following intravenous injection with an anti-mouse CD45 antibody (Supplementary Fig. 12). Then, lungs were cut into small pieces and placed in 2.5 ml of CO$_2$ Independent Medium containing 0.1 mg/ml DNase I and 0.1 mg/ml Liberase TL in C tubes (Miltenyi Biotec). After mechanical dissociation with a gentleMACS dissociator, samples were incubated with shaking at 37 °C for 30 min and finally processed again with the gentleMACS. The cell suspension was then filtered and mononuclear cells recovered from a Percoll gradient (GE Healthcare) from 40% to 75% interface.

**Flow cytometry analysis and Tetramer based cell enrichment**. For murine samples, single cell suspensions were preincubated with rat anti-mouse CD16/CD32 antibody (clone 2.4G2 produced in-house) for 15 min at 4 °C to block non-specific binding to Fcγ receptor. Staining was performed in PBS 0.5% BSA 2 mM EDTA with combinations of monoclonal antibodies (Supplementary Data 4). For human samples, PBMCs were incubated with FcR-blocking reagent (Miltenyi biotec) before cell surface staining (Supplementary Data 4). FOXP3 transcription factor staining buffer set (ebioscience) was used for intracellular staining. 4′,

6-diamidino-2-phenylindole (DAPI, Roche) or Aqua Dead Cell Stain Kit (Life Technologies) were used to exclude dead cells according to the absence or presence of a fixation step, respectively.

For tetramer cell enrichment, single cell suspensions were first stained with a hamster anti-mouse TCRβ antibody. Cells were then incubated for 1 h at 37 °C with allophycocyanin-DBY:I-A$^b$ tetramer (kindly provided by NIH myristate core facility, Emory University) at a final concentration of 10 μg/ml. Samples were then magnetically enriched. FACS data were acquired using an LSR Fortessa flow cytometer (BD Biosciences) and analyzed using FlowJo (version 10 ×, Tree Star).

**Cytokine production and suppression assay**. For cytokine production cell suspensions were seeded at $1 \times 10^6$/ml in complete RPMI-1640 (Life technologies) and cultured 4 h at 37 °C in 5% CO$_2$ with 20 ng/ml PMA (Phorbol myristate acetate, Sigma-Aldrich) and 1 μg/ml Ionomycin (Sigma-Aldrich). Alternatively, cell suspensions ($0.5 \times 10^6$/well) were restimulated for 24 h with CD3ε$^{-/-}$ female splenocytes ($0.5 \times 10^6$/well) pulsed with DBY peptide (10 nM). GolgiPlug (BD Bioscience) was added during the last 3 h. For suppression assays, FOXP3-GFP$^{Pos}$ host Tregs were sorted from tumors, TdLN or tumor ndLNs (pool of inguinal and mesenteric LNs) and were co-cultured with CFSE-labeled naive Marilyn cells (Tresponders, 2000/well) in the presence of CD3ε$^{-/-}$ female splenocytes (40,000/well) loaded with 2 nM of DBY peptide. Two-fold dilutions of Tregs were performed to obtain the different Tregs/Tresponders ratios. Cells were co-cultured for 3 days at 37 °C and proliferation assessed by flow cytometry based on CFSE dilution.

**mRNA extraction and microarrays**. Approximately $5 \times 10^4$ total Marilyn cells from 3 pools of 5 mice were purified by FACS from TdLN 7 days after transfer. Naive Marilyn cells and activated Marilyn cells from tumor-free B6 mice receiving male splenocytes into the f.p. or inoculated with the LV vector i.t. were used as controls. A second data set (3 pools of 5 mice per condition) was generated with 3000 CD4$^+$CD25$^{hi}$ host Tregs and FOXP3-GFP$^{Pos}$ Marilyn pTregs or FOXP3-GFP$^{Neg}$ Marilyn anergic cells purified from TdLN of mice bearing early or advanced tumors. FOXP3-GFP$^{Pos}$ Marilyn pTregs generated in tumor-free mice receiving DBY peptide i.t. and CD4$^+$CD25$^{hi}$ host Tregs from unmanipulated mice were used as controls. Total RNAs were extracted using RNeasy micro Kit (Qiagen) according to manufacturer's instructions. Total RNA concentration and RNA integrity was monitored by electrophoresis (Agilent Bioanalyzer, RNA 6000 Pico Assay). Gene expression analysis was conducted according to Affymetrix recommendations using Mouse Gene 2.1 ST arrays. Briefly, 1 ng (data set 1) or 100 pg (data set 2) of total RNA were processed in parallel with an external Mouse Universal Reference RNA to control robustness and reproducibility of enzymatic steps. Amplified and labeled molecules were monitored in order to hybridize arrays with 2.25 μg of labeled DNA. Raw data were generated and controlled with Expression console (Affymetrix) at the Institut Curie Genomic facility.

Human affymetrix data were generated from chCD4$^+$ T cells (CD3$^+$CD4$^+$CD127$^-$CD25$^-$) and Tconv cells (CD3$^+$CD4$^+$CD127$^+$CD25$^±$) purified by FACS from the blood of healthy donors ($n = 2$) or patients with metastatic uveal melanoma ($n = 3$), as previously described[11]. Total RNAs were extracted using RNeasy micro Kit (Qiagen). Samples were processed as described for the mouse samples and the hybridization was performed on human Gene 2.1 ST arrays.

**Statistical analysis**. Results were expressed as mean±SEM. Data analysis was performed on Prism software (GraphPad Prism v6) using unpaired student t-test or Mann–Whitney non-parametric test, as appropriate. Significances were indicated as follow *$p < 0.05$, **$p < 0.01$, ***$p < 0.001$, ns as non-significant.

**Transcriptome analysis**. Raw data files were processed: background corrected, normalized using the quantile method and analyzed using R with Bioconductor and packages Limma[48] and Oligo[49]. Genes were averaged using ProbeID and GeneName and transcripts were filtered using the refseq mRNA database. Principal component analyses on most differentially expressed genes, heatmaps and hierarchical clustering were performed using Qlucore Omics explorer 3.1. Significance threshold was defined for $q < 0.05$. Gene set enrichment analyses were performed, as previously described[50,51] using gene sets described previously (MSigDB database, Broad institute[32]).

**Data availability**. The affymetrix data supporting this study have been deposited in the GEO database under the accession numbers GSE113623, GSE113624, and GSE113625. The authors declare that the data supporting the findings of this study are available within the article and its Supplementary Information files, or are available upon reasonable requests to the authors.

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

## Acknowledgements

We thank Virginie Dangles-Marie, Celine Daviaud, Isabelle Grandjean, and Mickael Garcia, the mouse facility technicians and the flow cytometry core at Institut Curie. We thank the NIH tetramer core facility (Emory university) for providing DBY:I-A$^b$ tetramers. We thank Dr. Tyler Jacks and Dr. Michel Dupage for providing the KP mice, tools and methodological support. We thank Dr. Sebastian Amigorena and Dr. Claire Hivroz for discussions and reviewing the manuscript. R.A. was supported by Cancéropôle Ile de France, Fondation ARC pour la Recherche sur le Cancer and Fondation Trouver. This work was supported by the Institut National de la Santé et de la Recherche Médicale, Institut Curie, Fondation Trouver, Agence Nationale de la Recherche (ANR) (Labex DCBIOL and PACRI). O.L.'s group is supported by the "Equipe labellisée de la Ligue Contre le Cancer" program.

## Author contributions

R.A., C.S., E.P., and O.L. designed the research. R.A., H.F., S.L., C.S., E.B., I.P., V.P., J.D., M.S., A.D., and N.G.N. performed experiments and analyzed data. S.L. performed bioinformatics analysis of the transcriptome data. E.B. and V.P. performed the immunohistochemistry analyses. D.G. and B.S. provided reagents and contributed to

experimental design. H.F., C.S., M.S., D.G., B.S., and E.P. critically revised the manuscript. R.A., S.L., E.B., and O.L. wrote the paper.

## Additional information

**Competing interests:** The authors declare no competing interests.

