## [Peer Review File · Nature Communications]

Reviewers' comments:

Reviewer #1 (Tumor T response)(Remarks to the Author):

Alonso et al. utilize a TCR transgenic adoptive transfer system to investigate how an experimental lung tumour affects the activation of newly recruited naïve tumour-specific CD4 T cells. They describe that although Marylin T cells proliferated upon transfer in mice bearing tumours at different stages (early vs advanced), these cells lacked the ability to produce IFN-g upon in vitro restimulation. This is correlated with Treg conversion (<5%) and a phenotype described as 'anergic'. Consistent with the Treg phenotype, RNA sequencing data exposes several Treg-associated genes to be overexpressed in Marylin T cells from tumour-bearing mice. However, rather Marylin Tregs influencing these characteristics, the authors present data in support of the notion that host Tregs are involved. Together, with an additional approach utilizing a different priming site that induces better proliferative responses in tumour-bearing mice, the authors conclude that the tumour-draining lymph node is responsible for 'Treg induction and 'anergy' of newly-introduced tumour-specific CD4 T cells.

Major concerns:

Treg conversion of Marylin T cells: Differing between conditions, 3-5% of transferred Marylin T cells expressed FOXP3 upon transfer into tumour-bearing mice. While this appears to be a specific phenomenon that doesn't happen in the DBY+CpG controls, it is quite surprising that several Treg genes have come up in the bulk sequencing of the Marylin cells given this rather low frequency. Could it be that the fraction of Tregs is greater than what could be predicted from the FOXP3 staining?

Anergy: At this stage, anergy is being investigated purely by phenotype (FR4, CD73 and Ki67). No functional data is provided in support of this phenotype representing truly 'anergic' cells.

Functional consequences of these phenotypes: The authors show that depletion of host Treg cells in chimeric mice reverts the anergic phenotype of Marylin cells and results in stronger proliferation. They also show that priming in a different lymph node can have somewhat comparable effects on the transferred Marylin cells. However, the most important question is whether these changes have any functional consequences with regards to tumour control or are rather an epiphenomenon of a complex experimental tumour model that, for example, relies on additional delivery of certain miRs to work.

Minor concerns:

Endogenous response against LV: There were still about 30% of Marylin T cells diluting out CFSE altogether in the Mod-LV, suggesting that even in the Mod-LV there was substantial priming by transduced APC. This suggests that there might also be priming of an endogenous response. Accordingly, the development of the tumour should occur in the presence of a potential endogenous tumour-specific memory response. This is not investigated.

Circulating cells in the lung: The presence of undiluted marylin cells in the lung (Fig. 1B) suggest that the measurements rather reflect blood than lung. As mentioned by the authors, naïve cells are excluded from tissues from the lung. How do the authors exclude circulating cells in the lung preparations?

Anergic phenotype: Are FR4, CD73 and Ki67 expressed by the same cells? What is the expression pattern of FR4 and CD73 on proliferating cells?

CpG experiments: Young et al., PNAS 2007 showed that CpG blocks MHC II presentation by DC. Does this confound the interpretation of their results?

Reviewer #2 (T anti-tumor, T transcriptome)(Remarks to the Author):

In the current study by Alonso et al. the authors examined the ability of a genetically-induced tumor to promote anergy and peripheral Treg development among naïve CD4 T cells. The study examines mechanisms of peripheral tolerance during an immune response to tumor antigens and focuses specifically on the role of antigen presentation in the tumor-draining lymph node during differentiation of naïve CD4 T cells. The authors used the genetically-induced lung adenocarcinoma model and introduced the DBY antigen as a model neoantigen to track differentiation of the Marilyn transgenic CD4 T cells during the slow development of this tumor. The authors observed that the TCR transgenic cells adoptively transferred into mice at an early and advanced stage of tumor development were skewed toward Treg differentiation. This peripheral induction of Treg differentiation was documented at a gene expression, phenotypic and functional level. Generally, the figures are easy to follow and the data are clearly presented. Understanding the mechanisms that dictate the developmental fate of CD4 T cells responding to tumor are of significant importance and therefore this manuscript should be of interest to the readership of Nature Communications. A few minor comments are listed below to be addressed.

Minor comments

The development of Tregs in tertiary lymphoid tissues that develop in response to the genetic KRAS tumor model have been described (Immunity 2015 Vol 43. 579-590). While the authors of the current study highlight the development of Foxp3- Tregs arising in the draining lymph node due to differentiation from naïve T cells (peripheral Tregs), they should cite this previous study.

Line 721. "gently provided by Dr. T Jacks".

I imagine Dr. Jacks ever so delicately handing over the plasmid...

The authors likely meant generously.

Reviewer #3 (Treg, general T biology)(Remarks to the Author):

The authors describe a novel genetic tumor model expressing a conserved antigenic peptide. They focus on what happens to naïve CD4+ T cells that are specific for the antigen peptide when they are injected at two different stages of tumor development (early or late). The results are essentially the same--the majority of the naïve T cells proliferate, fail to develop into effector cells secreting IFN γ and are regarded as anergic based on the expression of so-called anergy markers (Fr4 and CD73). A small percentage of the T cells develop into Foxp3+ Treg. The major interesting finding is that the induction of both the anergic phenotype and the pTregs is dependent on the presence of antigen-specific host Tregs cells which are also present in the tumor microenvironment. There are a number of issues that should be addressed:

1. While the data presented is quite complete in terms of the phenotypic characterization of the transferred cells, no information is given as to whether these have any effect on the growth of the tumor. Tumor generation is prevented by depletion of the host Tregs, so this question should be relatively easy to address.
2. The most interesting finding in the paper is that the generation of the anergic cells and the pTregs is dependent on antigen-specific host Tregs. What are the kinetics of accumulation of the host Tregs during tumor development? Do the host Tregs express cell surface TGF β which has recently been implicated as playing a critical role in the induction of oral tolerance in a cell transfer model of oral tolerance (EJI 46: 1480, 2016). Do the host Tregs induce an anergic phenotype in host antigen-specific CD4+ cells?
3. The expression of Helios and Neuropilin on the pTregs is curious as both of these markers have

been claimed to be specific for Tregs.

4. Although high levels of FoxP3 and CD73 are newly discovered markers of anergic T cells, the meaning of the word "anergic" is much abused by immunologists. What happens if these cells are restimulated by their target peptide in vivo or in vitro? Do they die? Produce IL-2 and proliferate? Produce Th2 cytokines such as IL-10?

5. The in vitro suppressive activity of the antigen-specific Tregs isolated from the draining LN is not impressive. Are these sorted Tregs? they should inhibit responses almost completely at much higher Treg to responder ratios, eg. 1:16 or 1:32.

6. The data demonstrating that pTregs do not derive from anergic cells is correlative and not definitive. Cells with the anergic phenotype are only transferred from mice at a single time point and it is conceivable that anergic phenotype cells transferred at an earlier time point might differentiate into pTregs.

Point by point response to reviewers (Figures for the reviewer information compiled at the end)

Reviewer #1 (Tumor T response) (Remarks to the Author):

Alonso et al. utilize a TCR transgenic adoptive transfer system to investigate how an experimental lung tumour affects the activation of newly recruited naïve tumour-specific CD4 T cells. They describe that although Marilyn T cells proliferated upon transfer in mice bearing tumours at different stages (early vs advanced), these cells lacked the ability to produce IFN- γ upon *in vitro* restimulation. This is correlated with Treg conversion (<5%) and a phenotype described as 'anergic'. Consistent with the Treg phenotype, RNA sequencing data exposes several Treg-associated genes to be overexpressed in Marilyn T cells from tumour-bearing mice. However, rather Marilyn Tregs influencing these characteristics, the authors present data in support of the notion that host Tregs are involved. Together, with an additional approach utilizing a different priming site that induces better proliferative responses in tumour-bearing mice, the authors conclude that the tumour-draining lymph node is responsible for 'Treg induction and 'anergy' of newly-introduced tumour-specific CD4 T cells.

Major concerns:

1-Treg conversion of Marilyn T cells: Differing between conditions, 3-5% of transferred Marilyn T cells expressed FOXP3 upon transfer into tumour-bearing mice. While this appears to be a specific phenomenon that doesn't happen in the DBY+CpG controls, it is quite surprising that several Treg genes have come up in the bulk sequencing of the Marilyn cells given this rather low frequency. Could it be that the fraction of Tregs is greater than what could be predicted from the FOXP3 staining?

Answer

The Treg signature found in the bulk of activated Marilyn cells in the tumor-bearing mice is related to the high number of genes shared between the pTregs and the much more abundant anergic T-cells: We generated a second affymetrix data set from purified Marilyn pTregs (FOXP3-GFP^{Pos}) and anergic Marilyn cells (FOXP3-GFP^{Neg}) from TdLN. Principal Component Analysis (PCA) of these two data sets clearly showed that the bulk Marilyn cells from data set 1 cluster together with the pTregs and anergic cells from data set 2 and not with naive or activated cells from tumor-free mice (**Figure R1a**). Moreover, the bulk Marilyn from data set 1 in this PCA is next to the anergic group but clearly tends towards the pTreg population, corroborating the enrichment of the Treg gene signature in the bulk population. Moreover, a very low number of genes are specifically upregulated by anergic Marilyn cells (55 and 73 for early and advanced stage, respectively) when compared to pTregs and conversely a high number of genes are specific of pTregs (515 and 440 for early and advanced stage, respectively, including FOXP3 and IL2RA) when compared to the anergic cells (**Figure R1b**). Altogether these results show that the pTreg signature is wider than the anergic one and the genes are more strongly expressed, which explains that this signature is dominant in the bulk population despite pTregs being a minority.

2-Anergy: At this stage, anergy is being investigated purely by phenotype (FR4, CD73 and Ki67). No functional data is provided in support of this phenotype representing truly 'anergic' cells.

Answer

In the literature, the term "anergy" has different meanings as also pointed out by reviewer #3. The confusion stems from a lack of a positive definition. It is complex to show an absence of response: how many and which readouts should be used? The work of Mueller *et al* from Minneapolis enables a positive definition of anergic cells according to the expression of FR4 and CD73. Operationally, following this lead we chose to qualify all the cells expressing high levels of FR4 and CD73 as anergic cells. We acknowledge that the FR4^{hi}/CD73^{hi} CD4⁺ T-cell subset may have different properties according to the model used. The full functional characterization of the cells expressing high levels of these two markers is still a work in progress in the different systems. The current work is part of this process.

In our previous data, we already showed a lack of response of Marilyn cells in tumor-bearing mice compared to control with regard to proliferation, migration to the tumor site and IFN- γ production following PMA/Ionomycin restimulation. In **Figure 3c-3d of the main paper**, we now show the production of IFN- γ , IL-2 and IL-10 by Marilyn cells from tumor-bearing mice after *in vitro* restimulation with their cognate antigen. Activated Marilyn cells in the tumor context do not produce

IFN- γ or IL-10. Regarding IL-2, we did observe some level of secretion in the TdLN. However, the frequency of IL-2+ cells within the tumor was much lower than in the control. Importantly, in these experiments Marilyn T cells were studied 7 days after transfer, which may not have allowed the development of a full-blown anergic phenotype. Nevertheless, the data on Marilyn T-cells proliferation, IL-10 production, IFN- γ production and the decreased IL-2 production at the tumor site fit with the notion that FR4^{hi}CD73^{hi} cells are anergic in our model.

To better characterize the FR4^{hi}CD73^{hi} subset in our model, we performed another set of experiments (**Figure 4e-4h of the paper**) monitoring the fate of anergic cells after transfer into tumor-free mice left untreated or immunized with DBY+CpG. On the contrary to what we previously observed with anergic cells transferred into tumor-bearing mice (**Figure 4a-4c of the paper**), anergic Marilyn cells once reinfused in tumor free-mice regain their proliferative and migratory potential after *in vivo* immunization. Moreover, the expression of FR4 and CD73 dramatically decreased in these ex-anergic Marilyn cells. Altogether, these data show that 1) Marilyn cells activated in the tumor context are truly anergic, 2) the anergic phenotype is reversible and maintained by mechanisms operating in the TdLN and 3) FR4 and CD73 are robust markers of the anergic status in our setting.

3-Functional consequences of these phenotypes: The authors show that depletion of host Treg cells in chimeric mice reverts the anergic phenotype of Marilyn cells and results in stronger proliferation. They also show that priming in a different lymph node can have somewhat comparable effects on the transferred Marilyn cells. However, the most important question is whether these changes have any functional consequences with regards to tumour control or are rather an epiphenomenon of a complex experimental tumour model that, for example, relies on additional delivery of certain miRs to work.

Answer

We agree with the reviewer that it would have been great to show tumor rejection. However, for the reasons mentioned in the cover letter, we do not think that CD4⁺ T-cells are able to reject by themselves tumors expressing a cytoplasmic antigen. Our aim was not to study tumor rejection but was more fundamental: To characterize the CD4⁺ T-cell response against an antigen whose expression is restricted to a progressively developing tumor. Besides the many new phenomena we describe (1-indisputable generation of pTregs and not expansion of pre-existing contaminating Tregs; 2-generation of anergic T-cells in a tumoral context; 3-characterization of anergic cells at functional level; 4- reversibility of the anergic phenotype) that are interesting per se in our opinion, we think that our work is fully relevant to human tumor immunology since it opens up the possibility to reactivate these tumor antigen-specific anergic T-cells with a strong stimulation after depleting Tregs.

However, for the reviewer information we explored the relevance of our findings in a human cancer setting. We first defined the tumor-induced anergic gene signature using a new affymetrix data set consisting of anergic Marilyn cells purified from TdLN of mice bearing early or advanced tumors 7 days after transfer. As controls we used naive Marilyn cells and fully activated effector Marilyn cells from tumor-free mice receiving DBY+CpG i.t. To determine the specific gene signature of tumor-induced anergic Marilyn cells, we compared the transcriptomes of tumor-induced anergic Marilyn cells or effector Marilyn cells to naive Marilyn cells. 2064 genes were differentially expressed in effector Marilyn cells from tumor free mice as compared to 591 in and 715 in anergic cells of early and advanced tumors, respectively (**Figure R2a**). 187 genes specifically upregulated in the anergic cells from tumor conditions represent the tumor-induced anergic signature (**Figure R2b**). To determine whether the tumor-induced anergic gene signature, was also enriched in human CD4⁺ T-cells in cancer patients we took advantage of an unpublished transcriptome dataset from metastatic uveal melanoma patients and healthy donors generated in our lab. Our team recently showed that untreated metastatic uveal melanoma and breast cancer patients display an increased frequency and number of CD127⁺CD25⁺CD4⁺ T-cells in the blood as compared to healthy donor (**Figure R2c** taken from the article **Peguillet I et al. Cancer Res. 2014**). This population of chronically stimulated CD4⁺ T-cells (chCD4) is increased in cancer patients and displays some of the features found in the mouse anergic cells: reduced proliferation capacity and dampened cytokine secretion.

The human transcriptome data set includes purified chCD4⁺ T-cells and CD4 conventional T-cells from the blood of metastatic uveal melanoma patients and healthy donors. We found that the tumor-induced anergic gene mouse signature is significantly enriched in chCD4⁺ T-cells from tumor patients but not from healthy donors (**Figure R2d**). This indicates that our findings have some relevance to human oncology. This is now mentioned in the discussion of the paper.

Minor concerns:

4-Endogenous response against LV: There were still about 30% of Marilyn T cells diluting out CFSE altogether in the Mod-LV, suggesting that even in the Mod-LV there was substantial priming by transduced APC. This suggests that there might also be priming of an endogenous response. Accordingly, the development of the tumour should occur in the presence of a potential endogenous tumour-specific memory response. This is not investigated.

Answer

We agree this is an important point. In the previous version of the paper, the residual response that could be imputed to APC transduced by the Mod-LV or having captured antigen from Mod-LV transduced non-hematopoietic cells was evaluated using the high affinity TCR transgenic Marilyn cells that may not be representative of a polyclonal endogenous response. The DBY:I-A^b-Tetramer has been recently made available and allowed us to more precisely investigate this point. We found a very limited number of endogenous DBY-Tetramer+ cells in the mediastinal dLN 9 days after LV inoculation in regular B6 mice and these cells virtually disappeared within a week (**Figure R3a-R3b**). This early response was accompanied by a limited recirculation to the lung that did not increase over time. Of note, we did not observe any FOXP3 expression among DBY-Tetramer+ cells (**Figure R3c**). These results show that the APC having captured antigen from transduced cells or expressing residual antigen in the absence of complete silencing by the miRNA generated inefficient priming that rapidly vanishes without giving rise to memory or Tregs.

5-Circulating cells in the lung: The presence of undiluted Marilyn cells in the lung (Fig. 1B) suggests that the measurements rather reflect blood than lung. As mentioned by the authors, naïve cells are excluded from tissues from the lung. How do the authors exclude circulating cells in the lung preparations?

Answer

We agree that a very small number of undivided Marilyn T-cells are found despite lung perfusion with PBS twice before sample harvesting. These undivided T-cells could be related to tertiary lymphoid structure (although we don't observe such structures by histology) or incomplete washing despite our effort. To further determine the nature of these undivided cells, we performed intravascular staining by harvesting the lungs 3 minutes after intravenous injection of an anti-mouse CD45 antibody (CD45iv). As expected 100% of blood Marilyn cells were stained by the CD45iv antibody while no staining was observed in the tumor draining-lymph node (**Figure R4a**). In the lung around 10% of Marilyn cells were stained by the anti-CD45iv antibody showing a very limited contamination with blood after our washing process. To evaluate whether undivided Marilyn cells in the lung are circulatory, we quantified the frequency of CD45iv+ cells among undivided and divided Marilyn cells in the lung. Circulating Marilyn cells were found in both the undivided and divided compartments. Nevertheless, most of the undivided Marilyn cells were not stained with the CD45iv antibody (**Figure R4b-R4c**). Overall these results show that after extensive washing only very few circulating cells remain in the lung. The lung parenchyma contains a small but detectable number of undivided Marilyn cells. The exact location of these cells in the tumor is unclear at the moment.

6-Anergic phenotype: Are FR4, CD73 and Ki67 expressed by the same cells? What is the expression pattern of FR4 and CD73 on proliferating cells?

Answer

These data were already shown in **Figure 3c-3d** of the paper where a very low proliferation (low Ki67 expression) was observed on FR4^{hi}CD73^{hi} anergic cells contrasting with a high proliferation among effector Marilyn cells (FR4^{low}CD73^{low}) in the tumor-draining lymph node.

7-CpG experiments: Young et al., PNAS 2007 showed that CpG blocks MHC II presentation by DC. Does this confound the interpretation of their results?

Answer

The report of Young *et al.* showed that DC maturation by CpG treatment impairs the presentation of newly encountered MHCII-restricted antigens. Our data do not contradict these results as in our context CpG is administered in mice bearing DBY-expressing tumors and as a consequence DBY uptake and presentation by DCs have already occurred (**Figure 5 of the paper**).

Moreover, in the experiments in which we evaluated the impact of tumor antigen availability (**Figure 6 of the paper**), we injected DBY peptide that can directly be loaded on the membrane MHC-II molecules. Furthermore, CpG was administered simultaneously with DBY peptide, a condition that does not inhibit the response in the report of Young *et al.*

Reviewer #2 (T anti-tumor, T transcriptome)(Remarks to the Author):

In the current study by Alonso et al. the authors examined the ability of a genetically-induced tumor to promote anergy and peripheral Treg development among naïve CD4 T cells. The study examines mechanisms of peripheral tolerance during an immune response to tumor antigens and focuses specifically on the role of antigen presentation in the tumor-draining lymph node during differentiation of naïve CD4 T cells. The authors used the genetically-induced lung adenocarcinoma model and introduced the DBY antigen as a model neoantigen to track differentiation of the Marilyn transgenic CD4 T cells during the slow development of this tumor. The authors observed that the TCR transgenic cells adoptively transferred into mice at an early and advanced stage of tumor development were skewed toward Treg differentiation. This peripheral induction of Treg differentiation was documented at a gene expression, phenotypic and functional level. Generally, the figures are easy to follow and the data are clearly presented. Understanding the mechanisms that dictate the developmental fate of CD4 T cells responding to tumor are of significant importance and therefore this manuscript should be of interest to the readership of Nature Communications. A few minor comments are listed below to be addressed.

Minor comments

1-The development of Tregs in tertiary lymphoid tissues that develop in response to the genetic KRAS tumor model have been described (Immunity 2015 Vol 43. 579-590). While the authors of the current study highlight the development of Foxp3- Tregs arising in the draining lymph node due to differentiation from naïve T cells (peripheral Tregs), they should cite this previous study.

Answer

This reference is now cited (ref 30). In our study we also investigated the presence of TLS at the tumor site. We searched for lymphoid aggregates of >60 cells composed of B220⁺ and CD3⁺ cells that also present HEV structures according to PNAAd staining (**Figure R5a**). Cellular aggregates of variable size displaying well-defined T-cell and B-cell zones were found in the lung of tumor bearing mice (**Figure R5a-II to R5a-III**). However, the absence of PNAAd staining excludes the classification of these intrapulmonary aggregates as bona fide TLS (**Figure R5a-IV to R5a-V**). Neither the number nor the size of this aggregates increased with the tumor stage (**Figure R5b-R5c**). We only found TLS like structures in few mice at advanced stage in invasive tumor foci in the parietal pleura and the mediastinal space (**Figure R5a-VI to R5a-VII**)

2-Line 721. "gently provided by Dr. T Jacks".
I imagine Dr. Jacks ever so delicately handing over the plasmid...
The authors likely meant generously.

Answer

The correction has been done.

Reviewer #3 (Treg, general T biology)(Remarks to the Author):

The authors describe a novel genetic tumor model expressing a consecrated antigenic peptide. They focus of what happens to naïve CD4⁺ T cells that are specific for the antigen peptide when they are injected a two different stages of tumor development (early or late). The results are essentially the same-the majority of the naïve T cells proliferate, fail to develop into effector cells secreting IFN γ and are regarded as anergic based on the expression of so-called anergy markers (FR4 and CD73). A small percentage of the T cells develop into Foxp3⁺ Treg. The major interesting finding is that the induction of both the anergic phenotype and the pTregs is dependent on the presence of antigen-specific host Tregs cells which are also present in the tumor microenvironment. There a number of issues that should be addressed:

1. While the data presented is quite complete in terms of the phenotypic characterization of the transferred cells, no information is given as to whether these have any effect on the growth of the tumor. Their generation is prevented by depletion of the host Tregs, so this question should be relatively easy to address.

Answer

As already mentioned in the cover letter and in response to reviewer #1, we don't think that this experiment is relevant and feasible and would be really informative in the absence of positive control (we don't know how to induce a tumor rejection in this pure CD4⁺ T-cell model for a non-secreted antigen). It should be stressed that depleting Tregs can only be done for a short time, which is certainly not sufficient to observe tumor rejection in this model. In addition, it would require to make another batch of bone marrow chimera between FOXP3-DTR bone marrow and KP-RAG mice, wait for the reconstitution, induce the tumor, wait another 3-4 months for tumor growth and treat by DT for a week (one cannot do more, otherwise the mice die) and monitor tumor growth for another 3-4 months with luminometry which is a very imprecise method. Moreover, since the model we are using is asynchronous, as soon as the Treg depletion would have been stopped to keep the mouse alive, the host Tregs would rapidly reconstitute and prevent the naive Marilyn T-cells newly arriving in the TdLN to be efficiently primed.

2. The most interesting finding in the paper is that the generations of the anergic cells and the pTregs is dependent on antigen-specific host Tregs. What are the kinetics of accumulation of the host Tregs during tumor development? Do the host Tregs express cell surface TGF β which has recently been implicated as playing a critical role in the induction of oral tolerance in a cell transfer model of oral tolerance (EJI 46: 1480, 2016). Do the host Tregs induce an anergic phenotype in host antigen-specific CD4⁺ cells?

Answer

In the previous version of the article, we showed that the frequency of total host Tregs was higher in the TdLN and in the lung of tumor-bearing mice. For the reviewer information (**Figure R6a-R6b**), we evaluated the number of endogenous DBY-specific Tregs using DBY tetramer in tumor-free mice and in mice bearing early or advanced tumors. Host DBY:I-A^b-Tet⁺ Tregs were undetectable in the mediastinal lymph node of tumor-free mice whereas they were already present at early stage of tumor growth and subsequently accumulated during tumor development. Regarding the tumor site, endogenous DBY-specific Tregs were found since the early stage without change in number at later time points.

As we already demonstrated that the TdLN plays a major role in the induction of anergy/regulatory tumor-specific CD4⁺ T-cells, we evaluated the expression of LAP/GARP by host Tregs in the TdLN. In line with previous reports showing that LAP can be expressed on the membrane of activated Tregs (Andersson J *et al.* JEM 2008), we found an increased expression of cell surface LAP in total host Tregs of tumor-bearing mice. GARP was also expressed by host Tregs at a similar level in tumor-free and tumor-bearing mice (**figure R6c-R6d**). To formally assess the potential role of TGF- β , we administered every 3 days an anti-mouse TGF- β neutralizing antibody to mice bearing tumors starting 3 days before Marilyn cell transfer (**Figure R6e**). We did not observe any change in the number of Marilyn cells in the TdLN and in the lung. The treatment did not reduce the conversion of Marilyn cells into Tregs or the induction of anergy (**Figure R6f-R6h**). Therefore, TGF- β signaling is probably not involved alone in these processes.

We agree with the reviewer: "Do the host Tregs induce an anergic phenotype in host antigen-specific CD4⁺ T-cells?" it is an interesting question. However, we don't see how we could perform this experiment with the current technology. Tracking the conversion of naive polyclonal CD4⁺ T-cells into anergic cells in tumor-bearing mice would require specifically depletion of anergic T-cells with or without further depletion of Tregs. At the moment, only adoptive transfer of congenic marked TCR monospecific T-cells is possible to track naive cells becoming Tregs or anergic.

3. The expression of Helios and Neuropilin on the pTregs is curious as both of these markers have been claimed to be specific for Tregs.

Answer

Helios and Neuropilin-1 (Nrp-1) have been proposed as specific markers of thymus derived Tregs (tTregs) allowing distinction of tTregs and pTregs. However, others have argued that Helios can be expressed upon T-cell activation and proliferation or following Treg differentiation from

conventional T-cells *in vitro* and *in vivo* (Akimova T *et al.* PLoS One 2011 & Gottschalk R.A *et al.* J Immunol 2012). Regarding Nrp-1, although its expression has been used to distinguish pTregs and tTregs in the periphery of unmanipulated mice, pTregs can express Nrp-1 in inflamed tissues (Weiss J.M *et al.* J.Exp Med 2012). Moreover, a recent report using genetically modified mouse strains that favor either pTreg or tTreg generation showed that Helios and Nrp-1 do not unequivocally identify Treg clones of thymic or peripheral origin (Szurek E *et al.* PLoS One 2015). Thus, our results are consistent with the published literature.

4. Although high levels of FR4 and CD73 are newly discovered markers of anergic T cells, the meaning of the word "anergic" is much abused by immunologists. What happens if these cells are restimulated by their target peptide *in vivo* or *in vitro*? Do they die? Produce IL-2 and proliferate? Produce Th2 cytokines such as IL-10?

Answer

See answer to question #2 of reviewer #1

5. The *in vitro* suppressive activity of the antigen-specific Tregs isolated from the draining LN is not impressive. Are these sorted Tregs? they should inhibit responses almost completely at much higher Treg to responder ratios, eg. 1:16 or 1:32.

Answer

In the experiment assessing the suppressive activity of host Tregs (**Figure 7h of the paper**), Tregs were sorted according to FOXP3-GFP/CD25 expression without enrichment for DBY specificity. In our co-culture setup, the activation signal is brought by APC loaded with the DBY peptide and not by anti-CD3 antibody. Hence, in our setting only DBY-specific Tregs are activated. These Tregs only represent ~0.4% of total host Treg population which explains the relatively low suppression observed as compared with what would have been obtained if all Tregs were activated with an anti-CD3 antibody. Considering the low number of Tregs involved, the suppression we observed is in fact quite impressive.

6. The data demonstrating that pTregs do not derive from anergic cells is correlative and not definitive. Cells with the anergic phenotype are only transferred from mice at a single time point and it is conceivable that anergic phenotype cells transferred at an earlier time point might differentiate into pTregs.

Answer

We agree with the reviewer that the discussion about the filiation between pTregs and anergic cells relies only on correlative data and is not definitive as already stated in the previous version of the manuscript (we used the word "tentative filiation consideration"). Still the data are quite demonstrative as the pattern of CFSE dilution 7 days after the transfer of naïve Marilyn cells is so different between anergic, effector memory (**Figure 3e of the paper**) and pTregs (**Figure S4e of the paper**) that it excludes the possibility of pTregs deriving from anergic cells while it suggests that anergic cells derive from effector/memory cells. On the other hand, pTregs becoming anergic cannot be excluded.

It should be stressed that the system is asynchronous with continuous entry of new naïve Marilyn T-cells in the TdLN where priming occurs. Day 7 is already a quite early time point considering the 3-6 months tumor growth phase. Isolating anergic T-cells at an earlier time point would be technically very challenging given the low number of cells that could be isolated.

In answer to question #2 of reviewer #1, we already mentioned the transfer experiment in which anergic Marilyn cells transferred into tumor-bearing hosts remain anergic without any becoming pTregs whereas the transfer into tumor-free mice led to a loss of the anergic phenotype.

Figure R1

Figure R1: (a) PCA of bulk Marilyn cells and purified pTregs or anergic Marilyn cells from TdLN alongside with controls ($q < 0.05$). **(b)** Volcano plots comparing Marilyn pTregs and anergic cells with associated gene signature.

Figure R2

Figure R2: (a) Volcano plots representing the q value against fold-change gene expression for FACS-purified anergic or effector Marilyn cells versus naïve Marilyn cells. Anergic Marilyn cells were purified from TdLN of mice bearing early or advanced tumors. Effector Marilyn cells were purified from tumor-free mice receiving DBY+CpG i.t. Upregulated or downregulated genes (Fold change>1.5, q<0.05) are highlighted in red and green, respectively. **(b)** Venn-diagram of genes upregulated in anergic and effector Marilyn cells. In red is depicted the tumor-induced anergic signature. **(c)** Data from *Peguiet I et al. Cancer Res. 2014*. Increased frequency of CD127⁺CD25⁻CD4⁺CD3⁺ T-cells in the blood of patients with different type of cancer (healthy donor= HD, metastatic uveal melanoma=mUM, breast cancer=BC). The expression of CD127 and CD25 defines three subsets among CD4⁺ T-cells in human peripheral blood leucocytes. **(d)** Volcano plot comparing the transcriptomes of CD127⁺CD25⁻CD4⁺CD3⁺ T-cells to the one of CD4⁺ conventional (CD4conv) T-cells from healthy donors or metastatic uveal melanoma. In red is the anergic signature defined in **(b)**. The enrichment p value for each signature is shown.

Figure R3

Figure R3: (a-b) Number of activated DBY-specific endogenous CD4⁺ T-cells in the mediastinal draining-lymph node LN and the lung of regular B6 mice administrated with the Mod-LV. **(a)** Representative plots following DBY:I-A^b tetramer (Tet)-based cell enrichment of cell suspensions at different time point after LV inoculation. Total numbers of activated CD44^{hi}DBY:I-A^b-specific CD4⁺ T-cells are shown. **(b)** Quantification. Each point represents a pool of 2 mice. **(c)** FOXP3 expression by endogenous CD44^{hi}DBY:I-A^b-specific CD4⁺ T-cells 9 days after LV inoculation. One experiment out of two is shown.

Figure R4

Figure R4: Tumor-bearing mice were injected intravenously with 3 μg of anti-CD45 antibody (CD45iv). Three minutes later the lung were perfused and harvested. **(a)** Frequency of CD45iv+ Marilyn cells in different locations. **(b)** Representative plots of CD45iv staining in undivided and divided lung Marilyn CD4⁺ T-cells. **(c)** Quantification.

Figure R5

Figure R5: (a) CD3/B220 co-staining and PNAd staining in the lung of tumor-free or tumor-bearing mice at different stages of tumor growth. **(a-II & a-III)** Lymphoid aggregates in the lung of tumor-bearing mice are associated to vascular (V), bronchial or tumor structures (T). **(a-IV & a-V)** Absent of PNAd expression in the intrapulmonary lymphoid aggregates. **(a-VI)** In advanced tumor stage, the lymphoid aggregates present in the adenocarcinoma invading the chest wall **(a-VII)** display PNAd expression. **(b)** Number of lymphoid aggregates containing 60 or more CD3^+ T-cells and B220^+ B-cells at early ($n=8$) and advanced stage ($n=8$) in the whole lung of tumor-bearing mice. **(c)** Normalized lymphoid aggregate surface corresponds to the ratio of total lymphoid aggregate surface/total lung surface. All panels are at $\times 200$ magnification. Scale bar= $50 \mu\text{m}$.

Figure R6

Figure R6: (a) Representative plots of FOXP3 staining among CD44^{hi} host CD4⁺ T-cells following DBY:I-A^b tetramer (Tet)-based cell enrichment of cell suspensions from mice bearing early or advanced tumors. Total numbers of FOXP3⁺ among DBY:I-A^b-specific host CD4⁺ T-cells are shown. (b) Quantification. Pooled data of two independent experiments ***p<0.001 Mann-Whitney U test. (c) Representative plots of LAP and GARP cell surface expression in host Tregs. (d) Quantification. (e) Experimental schedule of the anti-TGF- β treatment in tumor-bearing mice. (f) Number of Marilyn cells (g) frequency of Marilyn pTregs and (h) frequency of anergic Marilyn cells in the TdLN and in the lung of tumor-bearing mice left untreated or treated with anti-TGF- β neutralizing antibody. One representative experiment out of two is depicted.

REVIEWERS' COMMENTS:

Reviewer #1 (Remarks to the Author):

I fully appreciate that studies cannot answer all questions and often leave open more questions than they answer. It is also sometimes necessary to generate artificial models of disease to investigate basic phenomena that are relevant for that disease. Accordingly, I have looked at the model created here favourably and have been impressed by the technical skills and hard work that have gone into its generation.

However, being unable to demonstrate that the central observation of this study (i.e. differentiation of tumour-specific CD4 T cells towards an anergic and/or pTreg phenotype) has any bearing on the disease that their model seeks to understand leaves me with two conclusions. Either the model doesn't appropriately mirror functional aspects relevant for the disease or the phenomenon observed has no relevance to the disease. The human data discussed suggest that it might be the first option, but unfortunately the data as it currently stands doesn't exclude option two.

Reviewer #2 (Remarks to the Author):

The authors have adequately addressed this reviewers previous comments.

Reviewer #3 (Remarks to the Author):

None